# Slice, Dice, and Optimize: Measuring the Dimension of Neural Network Class Manifolds

## Abstract

Deep neural network classifiers naturally partition input space into regions belonging to different classes. The geometry of these class manifolds (CMs) is widely studied and is intimately related to model performance; for example, the margin is defined via boundaries between these CMs. We present a simple technique to estimate the effective dimension of CMs as well as boundaries between multiple CMs, by computing their intersection with random affine subspaces of varying dimension. We provide a theory for the technique and verify that our theoretical predictions agree with measurements on real neural networks. Through extensive experiments, we leverage this method to show deep connections between the geometry of CMs, generalization, and robustness. In particular we investigate how CM dimension depends on 1) the dataset, 2) architecture, 3) random initialization, 4) stage of training, 5) class, 6) ensemble size, 7) label randomization, 8) training set size, and 9) model robustness to data corruption. Together a picture emerges that well-performing, robust models have higher dimensional CMs than worse performing models. Moreover, we offer a unique perspective on ensembling via intersections of CMs. Our core code is available on Github.

## 1 Introduction

Training neural networks to classify data is a ubiquitous and classic problem in deep learning. In $K$-way classification, trained networks naturally partition the space of inputs into $K$ regions, $S_k \subset R^D$,

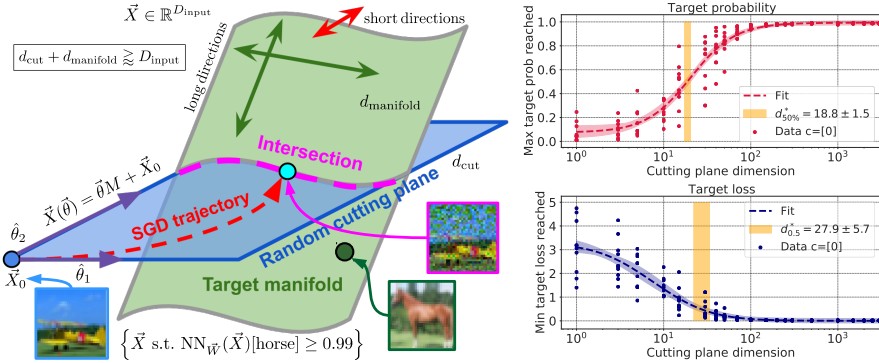

Figure 1: An illustration of finding a point in the intersection between a random cutting plane of dimension $d_{\mathrm{cut}}$ and a high-confidence manifold of effective dimension $d_{\mathrm{manifold}}$. If the $d_{\mathrm{cut}} \gtrsim D_{\mathrm{input}} - d_{\mathrm{manifold}}$, there likely exists an intersection between the two. We use optimization from a random point (image) $\vec{X}_0$ on the $d_{\mathrm{cut}}$ affine subspace to find a point in the intersection using gradient descent. The panels on the right show an example of the dependence of the probability and loss at the optimized point based on the $d_{\mathrm{cut}}$. The higher dimensional the cut, the less constrained the available images $\vec{X}$ are, and the more likely we are to find one of high class confidence.

containing points that the network confidently predict have class $k$. We call these regions *class manifolds* (CMs) of the neural network. In this paper, we analyze the high-dimensional geometry of these CMs, focusing primarily on their *effective dimensionality*.

To estimate the dimension of these CMs, we employ optimization on random $d$-dimensional sections of inputs space to beat the curse of dimensionality (Bellman, 1957) in order to seek out high-confidence regions that would be unlikely to be discovered at random with other diagnostic techniques. Through a theoretical analysis of high-dimensional geometry we link the success of such constrained optimization to the dimension of the target CM. Also, through extensive experiments, we leverage this method to show deep connections between the geometry of CMs, generalization, and robustness. In particular we investigate how CM dimension depends on 1) the dataset, 2) architecture, 3) random initialization, 4) stage of training, 5) class, 6) ensemble size, 7) training set size, and 8) model robustness to data corruption. Together a picture emerges that well-performing, robust, models have CMs that have higher dimension than inferior models. Moreover, we offer a unique perspective on ensembling via intersections of CMs.

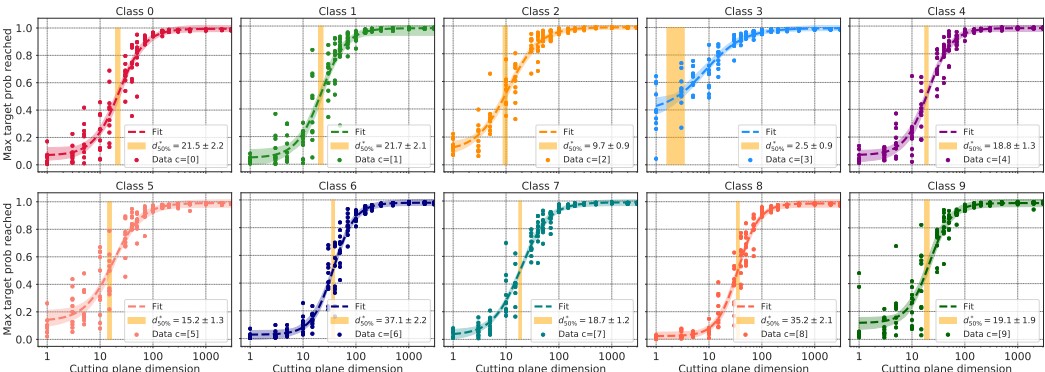

Figure 2: Maximum probability of single classes of CIFAR-10 reached on cutting planes of dimension $d$. The figure shows the dependence of the probability of a single class of CIFAR-10 (y-axes) reached on random cutting hyperplanes of different dimensions (x-axes). The results shown are for a well-trained ($> 90\%$ test accuracy) ResNet20v1 on CIFAR-10. Each dimension $d$ is repeated $10\times$ with random planes and offsets, and $d_{50\%}^*$ is extracted using a fit. The $d_{50\%}^* \ll 3072$, which implies that the class manifolds are surprisingly high dimensional ($3072 - d_{50\%}^*$). Indeed their dimensions *are all in excess* of 3000.

**Related work.** There has been a lot of research into understanding linear regions of neural networks, both trained and untrained. Montúfar et al. (2014) studied the number of linear regions in deep neural networks, Raghu et al. (2016) looked at their expressive power with depth, while Serra et al. (2017) tried to bound and count them. Hanin & Rolnick (2019b) showed that deep ReLU networks have surprisingly few activation patterns, and Hanin & Rolnick (2019a) did the same for the linear regions in the input space. The spectral properties of neural nets were studied in Rahaman et al. (2018), and the stiffness of the functional approximations defined through gradient alignment was coined in Fort et al. (2019). While revealing interesting aspects of neural network input space and activations space behavior, the methods used so far have not been able beat the curse of dimensionality – they have stayed local, and analyzed either one- or two-dimensional sections of input space.

The exploration of constrained optimization on random, $d$-dimensional planes in the weight space was employed successfully in Li et al. (2018) to estimate the intrinsic dimension of loss landscapes. Fort & Scherlis (2019) extended this analysis geometrically, and Fort & Jastrzebski (2019) used this and several other insights to build a geometric model of the low-loss basins weight-space basins.

Another closely related area of research concerns adversarial examples and robustness. Goodfellow et al. (2014) first noted that there exist points in input space very close to test examples that are mispredicted by neural networks, suggesting CMs of different classes can come very close to each other. Gilmer et al. (2018) showed that the existence of adversarial examples is related to the dimensionality of input space and the accuracy of the classifier. Ford et al. (2019) further link this interplay between dimensionality, generalization, and adversarial robustness to more general corruption robustness. In a similar spirit Salman et al. (2019) produce more robust models by convolving neural networks with Gaussian noise in input space. Whereas these studies are local, the techniques discussed in this paper are primarily concerned with global properties of CMs.

## 2 METHODS

We seek to determine the effective dimensionality of CMs. To that end, consider neural networks whose last layer is a softmax yielding normalized probabilities for a given input, $\vec{p}(\vec{X})$. Typically these networks have been trained using a cross entropy loss,

$$\mathcal{L}(p(\vec{X}), \hat{p}) = -\hat{p} \cdot \log[\vec{p}(\vec{X})]. \tag{1}$$

We define a class manifold for class $k$ as the pre-image, $S_k = \vec{p}^{-1}(\{p_k > p_{\text{threshold}}\})$. We seek to identify the effective dimension of $S_k$ by introducing a *the cutting plane method* (see also Fig. 1):

---

**The Cutting Plane Method:** For a neural network (NN) mapping input $\vec{X} \in \mathbb{R}^D$ into probabilities $\vec{p}(\vec{X}) \in \mathbb{R}^C$, take a random $d$-dimensional affine hyperplane defined by orthonormal basis vectors given by rows of $M \in \mathbb{R}^{d \times D}$ and a point $\vec{X}_0 \in \mathbb{R}^D$. Inputs in this hyperplane are parametrized by $\vec{\theta} \in \mathbb{R}^d$ as $\vec{X}(\vec{\theta}) = \vec{\theta}M + \vec{X}_0$. Given a target probability vector $\vec{p}_{\text{target}}$, we seek to optimize the cross entropy loss, $\mathcal{L}(\vec{p}(\vec{X}(\vec{\theta})), \vec{p}_{\text{target}})$ with respect to $\vec{\theta}$. This will identify points constrained to the affine subspace $(M, \vec{X}_0)$ that have probabilities as close as possible to $p_{\text{target}}$. We study the dependence of $\mathcal{L}_{\text{min}}$ and $\vec{p}_{\text{min}}$ (the loss and probability following optimization) on the dimension $d$. We show this analysis estimates the effective codimension of the preimage of $p_{\text{target}}$: $\{\vec{X} \text{ s.t. } \vec{p}(\vec{X}) = \vec{p}_{\text{target}}\}$.

---

We use Adam (Kingma & Ba, 2017) to minimize $\mathcal{L}$ with respect to $\vec{\theta}$, starting from $\vec{\theta}_0 = \vec{0}$, which corresponds to an initial *random* input $\vec{X}(\vec{\theta}_0) = \vec{X}_0$. Through optimization, we take $\vec{\theta}_0 \to \vec{\theta}_{\text{min}}$. The $\vec{\theta}_{\text{min}}$ defines an optimized input $\vec{X}_{\text{min}} = \vec{\theta}_{\text{min}}M + \vec{X}_0$ and corresponding output $\vec{p}_{\text{final}}(\vec{X})$ that is as close as possible to $\vec{p}_{\text{target}}$ while confining $\vec{X}$ to the random affine hyperplane defined by $(M, \vec{X}_0)$. We discuss the weak effect of sparsity of $M$ in 11.

The optimization thus starts with a tuple $(\text{NN}, d, M, \vec{X}_0)$ and maps it to the final probability vector $\vec{p}_{\text{final}}$ and the associated $\mathcal{L}_{\text{final}}$. By analyzing the dependence of $\vec{p}_{\text{final}}$ and $\mathcal{L}_{\text{final}}$ on the dimension $d$ of the random hyperplane we can access information about the effective dimensionality of the pre-image in input space of a region around $\vec{p}_{\text{target}}$ in output space (Fig. 1).

### 2.1 CLASS MANIFOLDS (CMS) AND MULTI-WAY CLASS BOUNDARY MANIFOLDS (CBMS)

There are several interesting choices of $\vec{p}_{\text{target}}$. Consider $\vec{p}_{\text{target}} = (0, 0, \ldots, 1, \ldots, 0)$, or a 1-hot vector on a single class $k$. Then the pre-image of $\vec{p}_{\text{target}}$ is the CM $S_k$. The cutting plane method enables us to estimate the effective co-dimension of $S_k$ by computing the dimension $d^*$ at which we reliably obtain a $p_{\text{final}}$ whose $k$'th component is close to 1 within some tolerance (Fig. 2).

The formulation in Equation 1 allows us to also study regions that lie in between classes. For example, by setting $\vec{p}_{\text{target}} = (\frac{1}{2}, \frac{1}{2}, 0, \ldots, 0)$, our optimization finds regions of input space that lie on a class boundary manifold (CBM) between classes 0 and 1. We can even find multi-way CBMs. For example, a three-way CBM between classes 0,1, and 2 corresponds to $\vec{p}_{\text{target}} = (\frac{1}{3}, \frac{1}{3}, \frac{1}{3}, 0, \ldots, 0)$. We can also study the region where all classes have equal probability by setting $\vec{p}_{\text{target}} = (\frac{1}{C}, \frac{1}{C}, \ldots, \frac{1}{C})$, where $C$ is the number of classes. Thus our method opens the door to study the intertwined geometry of multiple CMs and their boundaries. See Fig. 6 for results on multi-way CBMs.

### 2.2 EXTRACTING THE CRITICAL CUTTING PLANE DIMENSION AND CM CO-DIMENSION $d_{50\%}^*$

Given a particular class target vector $\vec{p}_{\text{target}}$ (i.e. $\vec{p}_{\text{target}} = (1, 0, \ldots, 0)$ corresponding to the CM $S_k$ with $k = 1$), we perform the cutting plane experiment multiple times for random $\vec{X}_0$ and $M$ for a sweep of different values of $d$. We obtain a final probability vectors $\vec{p}_{\text{min}}$ as a function of cutting plane dimension $d$, as shown in Figures 1 and 2. For CMs for a single class manifold $S_k$ we plot the $k$'th component $\vec{p}_{\text{min}}$. For small values of $d$ the affine cutting plane will not often intersect $S_k$ and $\vec{p}_{\text{min}}$ will be far from $\vec{p}_{\text{target}}$. For large dimensions, e.g. $d = D$, the subspace is now the full space of inputs, and we can therefore always find a point on the plane such that $\vec{p}_{\text{min}} \approx \vec{p}_{\text{target}}$. For intermediate values of $d$, the $k$'th component of $\vec{p}_{\text{min}}$ will gradually increase with $d$. To extract a

single cutting plane dimension from this data we 1) fit an empirical curve to the data (Equation 6; typically a good fit), 2) use the resulting distribution of fit parameters' mean and covariance to get a set of valid fitting functions, and 3) extracting the range of values of $d^*$ where these functions cross $p = 50\%$. We call this value $d^*_{50\%}$. (In some cases, we use other probability thresholds (25% and 75%) and we note that in the figures.) This cutting plane dimension is the *effective co-dimension* of the CM $S_k$. Thus the *effective dimension* of the CM $S_k$ is $D - d^*$ (as derived in Section 3).

## 3   A THEORY FOR ESTIMATING CM DIMENSION THROUGH CUTTING PLANES

Here we provide justification for the idea that $d^*_{50\%}$ obtained through the cutting plane method actually estimates the effective co-dimension of the CM $S_k$ (or of multiway CBMs), by analyzing what

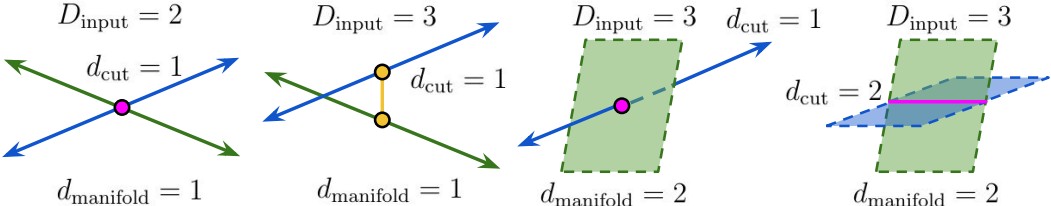

Figure 3: An illustration of the way two affine subspaces of dimensions $d_A$ and $d_B$ can intersect each other. If the dimension of the space $d_A + d_B \geq D$, the subspaces will likely intersect, while otherwise they typically do not intersect. If we know $D$ and $d_A$, we can use the existence of an intersection as a tool to bound $d_B \geq D - d_A$.

the cutting plane method would yield for a simple model of the CM as an affine hyperplane itself. To understand the relationship between the cutting plane dimension $d^*_{50\%}$ and the CM hyperplane dimension, we need to review the intersection theory of random affine hyperplanes.

### 3.1   INTERSECTION THEORY OF TWO RANDOM HYPERPLANES

For two hyperplanes (affine subspaces) of dimensions $d_A$ and $d_B$ that are randomly oriented and offset in an ambient vector space of dimension $D$, what is the condition on their dimensionalities that would make it *highly likely* that they intersect? The answer is

$$\boxed{d_A + d_B \geq D\,.} \tag{2}$$

In algebraic geometry, this statement is known as *dimension counting*, and is equivalent to the statement that the co-dimensions of hyperplanes are at most additive under intersection (Bourbaki, 1998) (recall that the co-dimension of a hyperplane of dimension $d$ in a space of ambient dimension $D$ is $D - d$). More precisely,

$$\max(\mathrm{codim}(A), \mathrm{codim}(B)) \leq \mathrm{codim}(A \cap B) \leq \mathrm{codim}(A) + \mathrm{codim}(B)\,. \tag{3}$$

Rewritten in dimensions, this means that $D - \dim(A \cap B)$ lies between $D - \min(d_A, d_B)$ and $D - (d_A + d_B)$. For the subspaces $A$ and $B$ intersecting transversally, which happens *generically*, the codimensions add exactly, satisfying the upper bound and therefore leading to Equation 2. An illustration of what such intersections can look like for $D = 2$ and $D = 3$ are shown in Figure 3.

In the cutting plane method, we control the dimension $d_A$ of a randomly chosen cutting hyperplane and use constrained optimization to find an intersection of this hyperplane with a CM in order to estimate its dimension $d_B$. The dimension $d_A = d^*_{50\%}$ where we can first reliably find a high confidence image of the target class will be the *codimension* of such a CM. The dimension of the CM will therefore be $d_B = D - d^*_{50\%}$. Based on the intersection theory of random hyperplanes, these results yield the exact dimension $d_B$ of the CM when it is a hyperplane of dimension $d_B$. However for more general CM's, the estimated $d_B = D - d^*_{50\%}$ is to be interpreted as an *effective dimension* of the CM and so $d^*_{50\%}$ itself is the effective co-dimension of the CM.

### 3.2   THEORY VS NUMERICAL EXPERIMENT

In Section A.2 we analytically derive the expected closest distance between two such affine subspaces. The result is the for $d_A + d_B \geq D$ it is 0 (they itersect),

while for $d_A + d_B < D$ the $\mathbb{E}\left[l(A, B)\right] \propto (\sqrt{D - d_A - d_B})/\sqrt{D}$. To compare this analytic result to reality, we ran a numerical experiment using automatic differentiation in JAX (Bradbury et al., 2018) where we generated random affine subspaces of different dimensions and measured their closest approach using optimization to locate the place. The numerical results presented in Figure 4 match the analytic predictions well.

### 3.3 LONG AND SHORT

DIRECTIONS − RELATING THEORY TO EXPERIMENTS

A hyperplane of dimension $d$ has $d$ infinitely extended directions and $D − d$ directions of size $0$. Among the $d$ extended dimensions, we can move arbitrarily far and still find ourselves in the hyperplane. Conversely moving even infinitesimally in the remaining $D−d$ directions will strictly lead to leaving the plane. In our experiments, this will be the case only approximately. In case of class manifolds, no directions will be infinitely thin or long, however, many will be *much thinner* than others. What we are measuring is therefore not strictly a dimension (the manifold will always have the full dimension $D$) but rather *effective dimension* (Vershynin, 2015). However, our hyperplane model works well in practice.

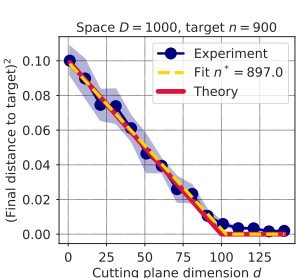

Figure 4: Distance between random affine subspaces − numerical experiments vs theory from Eq. 4. The figure compares the distance between two subspaces of dimensions $d$ & $n$ in a $D$-dimensional space. The numerical experiment in JAX is in blue, theory in red, and numerical fit in yellow.

## 4 EXPERIMENTS

We now describe experiments validating our random cutting plane method of identifying the dimension of CMs and connections between these measurements, generalization, and robustness. The details of the architectures, datasets and precise training procedures are detailed in the Appendix Section A.1. The majority of our experiments are done with a standard ResNet20v1 on CIFAR-10 and CIFAR-100.

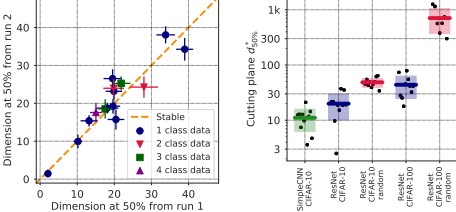

### 4.1 RE-INITIALIZATION

AND RE-TRAINING STABILITY

For our technique to have predictive power, cutting plane results should be stable under reinitialization and retraining of a model. Given a fully trained model, running a sweep over cutting plane dimension $d$ several times should produce consistent results. We verified that our

Figure 5: **Left panel**: Comparison of the cutting plane dimension needed to get $50\%$ of the target class ($d^*_{50\%}$) for two independently initialized and trained ResNets on CIFAR-10, showing the stability of our method. **Right panel**: Comparison of the $d^*_{50\%}$ for SimpleCNN and ResNet on CIFAR-10 and CIFAR-100 with real and *randomized* labels.

method is stable, as shown in Figure 5 where we compare the $d^*_{50\%}$ dimensions extracted from single class regions of CIFAR-10, as well several regions between 2, 3 and 4 classes. The results are consistent between the two runs.

### 4.2 SINGLE CLASS MANIFOLDS

The main object of interest for us are the high-confidence single class manifolds. We present our results for a well-trained ResNet20v1 on CIFAR-10 in Figure 2, and for CIFAR-100 in Figure 18. We also show results for a SimpleCNN on CIFAR-10 in Figure 17. The results show that the $d^*_{50\%}$ is very small compared to the dimension of the input, suggesting that the class manifold dimension is actually very high, close to the full 3072 dimensions for CIFAR-10. The connection between $d^*$ and the manifold dimension is derived in Section A.2.

### 4.3 CLASS BOUNDARY MANIFOLDS BETWEEN MULTIPLE CLASSES

As described in Section 2.1, our method allows us to study the dimensionality of boundary manifolds in between multiple classes as well. We show results for a well trained ResNet20v1 on CIFAR-10 ($> 91\%$ test accuracy) for several selected sets of classes in Figure 6. In particular, we look at the region in between all 10 classes, where the network is equally uncertain about all classes. There, we primarily focus on the loss (as described in Equation 1) in the bottom row of Figure 6

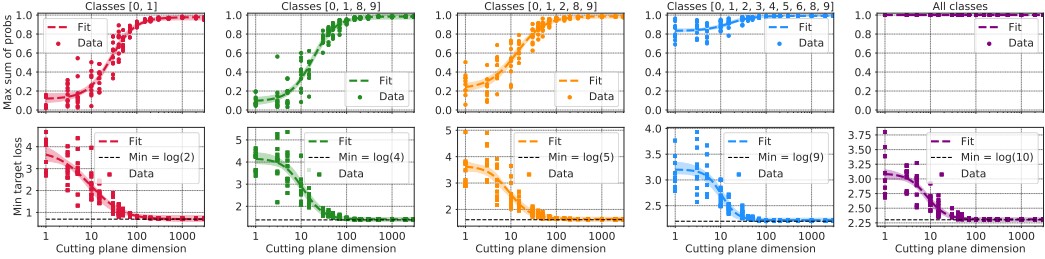

Figure 6: Maximum probability and minimum loss (y-axes) of in-between-classes regions of CIFAR-10 reached on cutting planes of dimension $d$ (x-axes). The results shown are for a well-trained ($> 90\%$ test accuracy) ResNet20v1 on CIFAR-10. Each dimension $d$ is repeated $10\times$ with random planes and offsets. The last column shows the results for the 10-class region where the network assigns equal probability to each class.

### 4.4 TRAINING ON RANDOM LABELS

Due to the structure of the training data and the neural network prior, we expect the learned class manifolds to inherit a lot of structure from both. To disentangle the role of the class label, we trained a ResNet20v1 on CIFAR-10 with randomly reshuffled labels. As shown in Zhang et al. (2017), we can reach $100\%$ training accuracy on random labels. However, as shown in Figure 5 and 12 the CMs learned in such way have a significantly higher $d_{50\%}^*$ and therefore smaller dimension. Since these models completely fail to generalize, this result is consistent with the hypothesis that generalization and class manifold dimension are intimately related.

### 4.5 THE EFFECT OF TRAINING SET SIZE

During the course of training, a neural network has to learn to partition the $D$-dimensional space of inputs into generalizable regions of high class confidence that contain both the training points (by training) and the test points (by generalization). To see the role of training set size, we repeated our cutting plane experiments for networks trained to $100\%$ training set accuracy on subsets of CIFAR-10 of size 250, 500, 1,500, 5,000, 15,000, and 50,000 images (=full training set) and added a final point where we used data augmentation on top of the full training set. The bigger the training set, the smaller the $d_{50\%}^*$, and therefore the larger the dimension of the CMs, as shown in Fig. 7. This trend held across all classes, and continued from the full training set to the training set and augmentation. Thus again, better generalization is associated with higher CM dimensionality. We hypothesize that the larger number of training points might allow the learned partitioning of the input space to connect previously disconnected and lower dimensional CMs through interpolation, thereby effectively increasing CM dimensions with training set size.

### 4.6 THE EFFECT OF DATA CORRUPTION

We measure the effect of cutting plane dimension on out-of-domain robustness of neural networks. Robustness to Gaussian noise was found to be a useful predictor for general robustness as well as adversarial robustness (Ford et al., 2019; Yin et al., 2019). For this reason, we first measure the robustness of Wide-ResNet models to Gaussian noise applied at test time, where noise is sampled from a Gaussian with 0.05 standard deviation for each pixel independently. Left panel of Fig. 8 shows the correlation between the cutting plane dimension and error due to Gaussian noise, calculated as the accuracy on corrupted data minus the accuracy on clean data. We see that the models with smaller cutting plane dimension, and therefore higher CM dimension, are more robust to this type of noise.

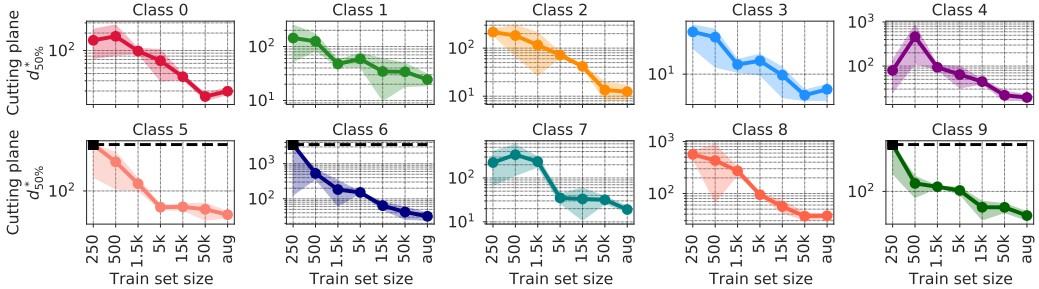

Figure 7: Comparison of the cutting plane dimension needed to get $50\%$ of the target class for ResNet trained to $100\%$ training accuracy on subsets of the training set of CIFAR-10 (mean and standard deviation of two networks). The bigger the training set, the smaller the $d^*_{50\%}$, meaning that its easier to find high confidence images, and that their manifolds increase in dimension. The trend continues with the addition of data augmentation (aug).

Next, we calculate the correlation between the cutting plane dimension and the accuracy on CIFAR-10-C (Hendrycks & Dietterich, 2018), which includes 15 different corruption types applied at test time (right panel of Fig. 8). These results together show that the cutting plane dimension of neural networks is correlated with their robustness to a variety of test-time distortions. Again, higher class CM dimension leads to better performance in terms of robustness.

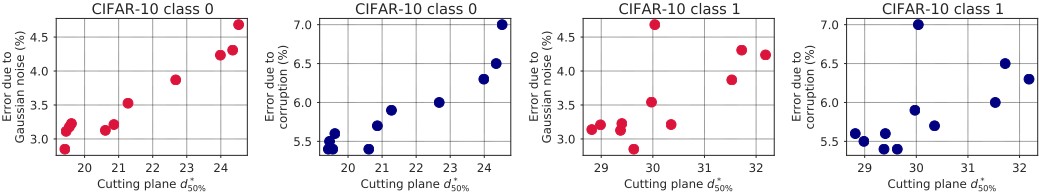

Figure 8: The effect of cutting plane dimension on model robustness to test-time distortions. The left panel shows the error due to Gaussian noise applied at test time vs. cutting plane dimension needed to get 50% of the points to target classes 0 and 1. The right panel shows the effect of the cutting plane dimension on error due to corruptions in CIFAR-10-C. Models with lower cutting plane dimension are more robust to both Gaussian noise and to distortions in the Common Corruptions Benchmark.

### 4.7 EVOLUTION OF DIMENSION WITH TRAINING

We study the effect of training on the high confidence manifolds in Figure 9. The early epochs are heavily influenced by the initialization. After a small amount of training, there seems to be an intermediate stage when it is very hard to find high confidence class manifolds ($d^*_{25\%}$ is high, and therefore the manifold dimension is low). Towards the end of training, $d^*_{25\%}$ goes down for all classes (detailed Figures 16, 18 and 17). The non-monotonic behavior of the dimension points towards something unusual happening in the intermediate stages of training, and it could be related to the host of phenomena pointing towards the high impact of early stages of training.

### 4.8 MODEL ENSEMBLING

We found that model ensembling (taking $N$ independently trained models, giving them the same input, and averaging their predicted probabilities) leads to class manifolds of lower dimension, as well as between-class regions of lower dimension. The bigger the ensemble, the lower the dimension, as shown in a summary plot in Figure 10. This is atypical, as all other methods of improving performance (e.g. larger training set, more training (towards the end)) correlated with higher dimensional CMs. This suggests that ensembles might be doing something geometrically distinct from the other methods. This could be related to the observation that, unlike other techniques, deep ensembles combine models from distinct loss landscape basins Fort et al. (2020).

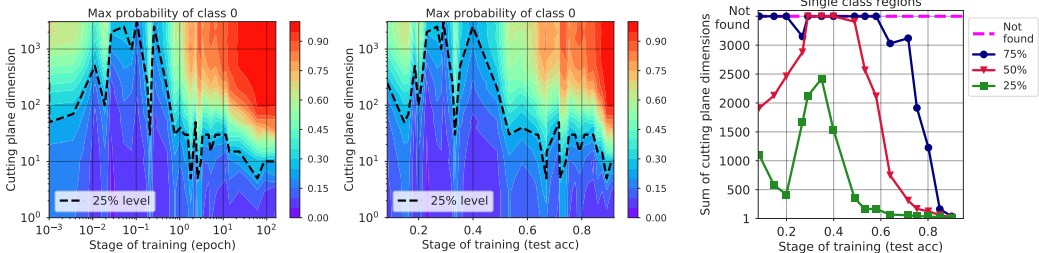

Figure 9: The effect of training on the dimension of cutting plane necessary to reach a particular probability. The two left panels show the maximum probability of class 0 reached for cutting planes of different dimensions (y-axes) for different stages of training of a ResNet20v1 on CIFAR-10 (x-axes). The probability 25% level is highlighted. The right panel shows $d_{25\%}^*$, $d_{50\%}^*$ and $d_{75\%}^*$ for the average of all single class regions. We can see an intermediate stage of training when high confidence regions become hard to find. Towards the end of training, the dimension of the manifolds grows. The breakdown by classes is shown in Figure 16.

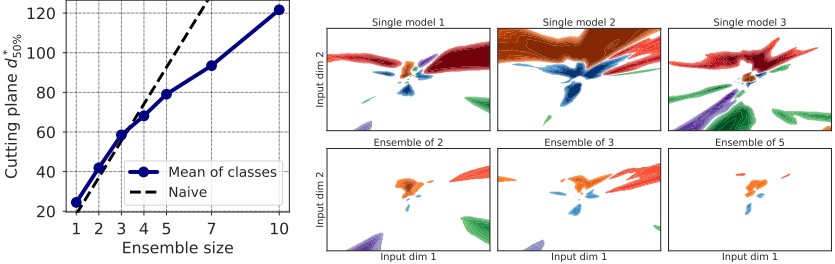

Figure 10: The effect of model ensembling on the dimension $d_{50\%}^*$ needed to reach 50% accuracy averaged over all CIFAR-10 classes (individual results in Fig. 13) for ResNet20 trained for 50 epochs. Across all classes, the larger the ensemble, the higher the $d_{50\%}^*$ and therefore the lower the class manifold dimension. A naive model of addition of codimensions between models is overlayed, showing a surprisingly good fit for small ensembles. The right panels show a section of the input space for 3 single models (top row) and 3 ensemble sizes (bottom). The colors indicate 4 different classes $> 50\%$. The elongated high-probability structures disappear with ensembling.

## 5 CONCLUSION

We propose a new method for estimating the dimension of both class manifolds and multi-way class boundary manifolds in the space of inputs for deep neural neural networks. To circumvent the curse of dimensionality, we use optimization constrained to randomly chosen affine subspaces (hyperplanes) of varying dimension. This allows us to extract the effective dimension of the class manifolds as well regions between classes. We study the manifold dimension as a function of 1) architecture, 2) dataset, 3) the amount of training, 4) dataset size, 5) data augmentation, 6) label randomization, 7) robustness to noise and perturbations, and 8) ensemble size. The ubiquitous correlation between higher class manifold dimension and better performance and robustness along the many axes tested points towards an intimate link between the geometry of the input space class partitioning and generalization. Ensembling, on the other hand, both increases performance and decreases the manifold dimension, and is the only technique amongst the ones we explored that does so, suggesting that its beneficial effects might be geometrically distinct from other ways of improving performance.

Overall the development of a full theory of deep learning poses a difficult intermediate theoretical problem: understanding the shape and evolution over training of complex nonlinear maps from high $D$-dimensional input spaces to high $C$-dimensional class probability spaces. By slicing, dicing and optimizing such high dimensional nonlinear maps using our cutting plane method, we hope our work opens the door to developing a more veridical geometric perspective on the nature of nonlinear maps learned by deep neural network classifiers.

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

# A   APPENDIX

## A.1   DETAILS OF NETWORKS, DATASETS AND TRAINING

In this paper we use two architectures: 1) `SimpleCNN`, which is a simple 4-layer CNN with 32, 64, 64 and 128 channels, ReLU activations and `maxpool` after each convolution, followed by a fully-connected layer, and 2) `ResNet20v1` as described in He et al. (2015) with batch normalization on (Ioffe & Szegedy, 2015). We use 5 datasets: MNIST (LeCun & Cortes, 2010), Fashion MNIST (Xiao et al., 2017), CIFAR-10 and CIFAR-100 (Krizhevsky et al.), and ImageNet Deng et al. (2009). The ResNet is trained for 200 epochs using SGD+Momentum at learning rate 0.1, dropping to 0.01 at epoch 80 and 0.001 at epoch 120. The $L_2$ norm regularization is $10^{-4}$. In one experiment, we use data augmentation as described in [1]. For our robustness experiments, we used the Wide-ResNet models (Zagoruyko & Komodakis, 2016) available in [2]. We trained 11 different sizes of Wide-ResNet models (WRN-28-2 to WRN-28-12) with AutoAugment. Each model was trained from 15 different random weight-initializations for better statistics. We used the following hyperparameters to train each model: a learning decay of 0.1, weight decay of 5e-4, cosine learning rate decay in 200 epochs, and AutoAugment (Cubuk et al., 2018) for data augmentation.

## A.2   DETAILED DERIVATION OF THE CLOSEST APPROACH OF TWO AFFINE SUBSPACES

Let us consider a situation in which in a $D$-dimensional space we have a randomly chosen $d$-dimensional affine subspace $A$ defined by a point $\vec{X}_0 \in \mathbb{R}^D$ and a set of $d$ orthonormal basis vectors $\{\hat{v}_i\}_{i=1}^d$ that we encapsulate into a matrix $M \in \mathbb{R}^{d \times D}$. Let us consider another random $n$-dimensional affine subspace $B$. Our task is to find a point $\vec{X}^* \in A$ that has the minimum $L_2$ distance to the subspace $B$, mathematically $\vec{X}^* = \mathrm{argmin}_{\vec{X} \in A} \left| \vec{X} - \mathrm{argmin}_{\vec{X}' \in B} \left| \vec{X} - \vec{X}' \right| \right|$. In words, we are looking for a point in the $d$-dimensional subspace $A$ that is as close as possible to its closest point in the $n$-dimensional subspace $B$. A point within the subspace $A$ is parametrized by a $d$-dimensional vector $\vec{\theta} \in \mathbb{R}^d$ by $\vec{X}(\theta) = \vec{\theta}M + \vec{X}_0 \in A$. This parametrization ensures that for all choices of $\vec{\theta}$ the resulting $\vec{X} \in A$.

Without loss of generality, let us consider the case where the $n$ basis vectors of the subspace $B$ are aligned with the dimensions $D - n, D - n + 1, \ldots, D$ of the coordinate system. Let us call the remaining axes $s = D - n$ the *short* directions of the subspace $B$. A distance from a point $\vec{X}$ to the subspace $B$ now depends only on its coordinates $1, 2, \ldots, s$. Therefore $l^2(\vec{X}, B) = \sum_{i=1}^s X_i^2$. This is the case because of our purposeful choice of coordinates.

Given that the only coordinates influencing the distance are the first $s$ values, let us, without loss of generality, consider a $\mathbb{R}^s$ subspace of the original $\mathbb{R}^D$ only including those. Then the distance between a point within the subspace $A$ parametrized by the vector $\vec{\theta}$ is $l^2(\vec{X}(\theta), B) = \left| \vec{\theta}M + \vec{X}_0 \right|^2$.

Given our restrictions, now the $\theta \in \mathbb{R}^d$, $M \in \mathbb{R}^{d \times s}$ and $\vec{X}_0 \in \mathbb{R}^d$. The distance $l$ attains its minimum for $\partial_{\vec{\theta}} l^2 = \left( \vec{\theta}M + \vec{X}_0 \right) M^T = \vec{0}$, producing the minimality condition $\vec{\theta}^* M = -\vec{X}_0$. There are now 3 cases:

**1. The overdetermined case,** $d > s$**.** In case $d > s = D - n$, the optimal $\theta^* = -\vec{X}_0 M^{-1}$ belongs to a ($d - s = d + n - D$)-dimensional family of solutions that attain 0 distance to the plane $B$. In this case the affine subspaces $A$ and $B$ intersect and share a ($d + n - D$)-dimensional intersection.

**2. A unique solution case,** $d = s$**.** In case of $d = s = D - n$, the solution is a unique $\theta^* = -\vec{X}_0 M^{-1}$. After plugging this back to the distance equation, we obtain $\vec{\theta}$ is $l^2(\vec{X}(\vec{\theta}^*), B) = \left| -\vec{X}_0 M^{-1} M + \vec{X}_0 \right|^2 = \left| -\vec{X}_0 + \vec{X}_0 \right|^2 = 0$. The square (in this case) matrix $M$ and its inverse $M^{-1}$ cancel each other out.

---

[1] https://github.com/keras-team/keras/blob/master/examples/cifar10_resnet.py

[2] https://github.com/tensorflow/models/tree/master/research/autoaugment

**3. An underdetermined case, $d < s$.** In case of $d < s$, there is generically no intersection between the subspaces. The inverse of $M$ is now the Moore-Penrose inverse $M^+$. Therefore the closest distance is $\vec{\theta}$ is $l^2(\vec{X}(\vec{\theta}^*), B) = \left| -\vec{X}_0 M^+ M + \vec{X}_0 \right|^2$. Before our restriction from $D \to s$ dimensions, the matrix $M$ consisted of $d$ $D$-dimensional, mutually orthogonal vectors of unit length each. We will consider these vectors to be component-wise random, each component with variance $1/\sqrt{D}$ to satisfy this condition on average. After restricting our space to $s$ dimensions, $M$'s vectors got reduced to $s$ components each, keeping their variance $1/\sqrt{D}$. They are still, in expectation, mutually orthogonal, however, their length got reduced to $\sqrt{s}/\sqrt{D}$. The (transpose) of the inverse $M^+$ consists of vectors of the same directions, with their lengths scaled up to $\sqrt{D}/\sqrt{s}$. That means that, in expectation, $MM^+$ is a diagonal matrix with $d$ diagonal components set to 1, and the remainder being 0. The matrix $(I - M^+ M)$ contains $(s - d)$ ones on its diagonal. The projection $|\vec{X}_0(I - M^+ M)|^2$ is therefore of the expected value of $|X_0|^2(s - d)^2/D$. The expected distance between the $d$-dimensional subspace $A$ and the $d$-dimensional subspace $B$ is, in expectation

$$\mathbb{E}\left[d(A, B)\right] \propto \begin{cases} \frac{\sqrt{D-n-d}}{\sqrt{D}} & n + d < D \,, \\ 0 & n + d \geq D \,. \end{cases} \tag{4}$$

We ran a numerical experiment using automatic differentiation in JAX (Bradbury et al., 2018) where we generated random affine subspaces of different dimensions and measured their closest approach using optimization to locate the place. The numerical results presented in Figure 4 match the analytic predictions in Equation 4 well.

## A.3 EMPIRICAL FIT FUNCTION

The empirical fit function we use to extract the critical dimension of the cutting hyperplane $d_{50\%}^*$ is shown in 6.

$$p(d; A, B, C, D) = A + \frac{B}{1 + \exp\left(-\log(d/C)/D\right)} \,. \tag{5}$$

It is a sigmoid function that depends logarithmically on the dimension $d$ and can be offset from $p = 0$ at for low $d$ and from $p = 1$ for high $d$. That is the case as sometimes the neural networks we analyzed would not have any regions of a particular class reaching all the way to $100\%$. In other cases, even optimization in a line $d = 1$ would be able to get to a $p > 10\%$ (for 10 class classification).

For fitting the loss $\mathcal{L}(d)$, we utilized the fact that the cross-entropy loss depends logarithmically on $p$, and therefore used

$$\mathcal{L}(d; A, B, C, D) = -\log\left[A + \frac{B}{1 + \exp\left(-\log(d/C)/D\right)}\right] \,. \tag{6}$$

In both cases $A$, $B$, $C$ and $D$ are free fit parameters. We used SciPy optimizer (Virtanen et al., 2020) to find the parameters and their covariance.

## A.4 CUTTING PLANE AXIS-ALIGNMENT – THE EFFECT OF SPARSITY

When choosing the matrix $M$ that defines the span of the subspace in which we are optimizing, we can choose to make the rows of $M$ sparse. On one end, each basis vectors might generically be non-zero in each of its components, while on the other end, a single non-zero element per basis vector is allowed. Geometrically, this corresponds to the alignment of the subspace with the axes (pixels and their channels for images) of the input space. Figure 11 shows the effect of the sparsity of $M$ on the resulting $d_{25\%}^*$, $d_{50\%}^*$, $d_{75\%}^*$ and $d_{90\%}^*$. The sparser the $M$, the higher the dimension needed to reliably reach the $25\%$, $50\%$, $75\%$, and $90\%$ class confidence region respectively. The effect of sparsity is visible, however, it is 1) not very significant (changing the dimension by a small part of the total $D = 3072$ for CIFAR-10), and 2) its effect disappears for even small amounts of non-zero elements in $M$.

## A.5 TRAINING ON RANDOMLY PERMUTED LABELS

For training on randomly permuted labels of the training set, we observe the critical dimension $d_{50\%}^*$ to rise significantly, meaning that a much higher dimensional cutting plane is needed to reliably

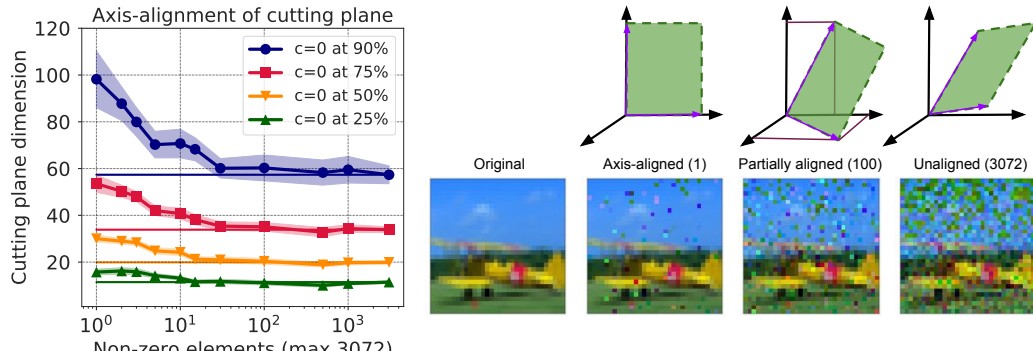

Figure 11: The effect of axis alignment of the cutting planes. The figure shows the cutting plane dimension necessary to reach 4 thresholds levels (the 4 data lines) of class 0 probability (y-axis) from a random starting point for a well trained ResNet20v1 on CIFAR-10. We vary the number of non-zero elements of the basis vectors of the random cutting plane (x-axis). For a small number of non-zero elements, single pixels are varied, while for a 3072 non-zero elements (the maximum value), all pixels are varied jointly. The axis-aligned random cuts require higher dimensions to hit the same accuracy regions of class 0.

intersect a class manifold. The breakdown by class for ResNet20v1 on CIFAR-10 and CIFAR-100 is shown in Figure 12. The comparison to semantically meaningful labels is shown in Figure 5.

## A.6 ENSEMBLING

The effect of ensembling on the critical dimension $d_{50\%}^*$ broken down by class is shown in Figure 13. For each of the 10 CIFAR-10 classes, the $d_{50\%}^*$ grows with ensemble size, which is unlike any other performance-improving techniques (such as data augmentation, and more training) we experimented with. A simple model predicting the resulting $d_{50\%}^*$ for an ensemble of size $n$ by assuming that the class manifold codimensions add works surprisingly well for small ensemble sizes in Figure 13. The intuition for why that might be the case is illustrated in Figure 10.

## A.7 ADDITIONAL CUTTING CURVES FOR CIFAR-10 AND CIFAR-100

Two additional detailed cutting plane results can be found in this subsection: SimpleCNN on CIFAR-10 in Figure 14, and ResNet20v1 on CIFAR-100 in Figure 15.

## A.8 DIMENSION AS A FUNCTION OF TRAINING STAGE

While Figure 9 shows the aggregate effect of training epoch on the the critical cutting plane dimension averaged over all single-class regions, the detailed per-class results can be found in Figure 16 for ResNet20v1 on CIFAR-10 (two indepdently initialized and trained models), in Figure 17 for SimpleCNN on CIFAR-10, and in Figure 18 for ResNet20v1 on CIFAR-100.

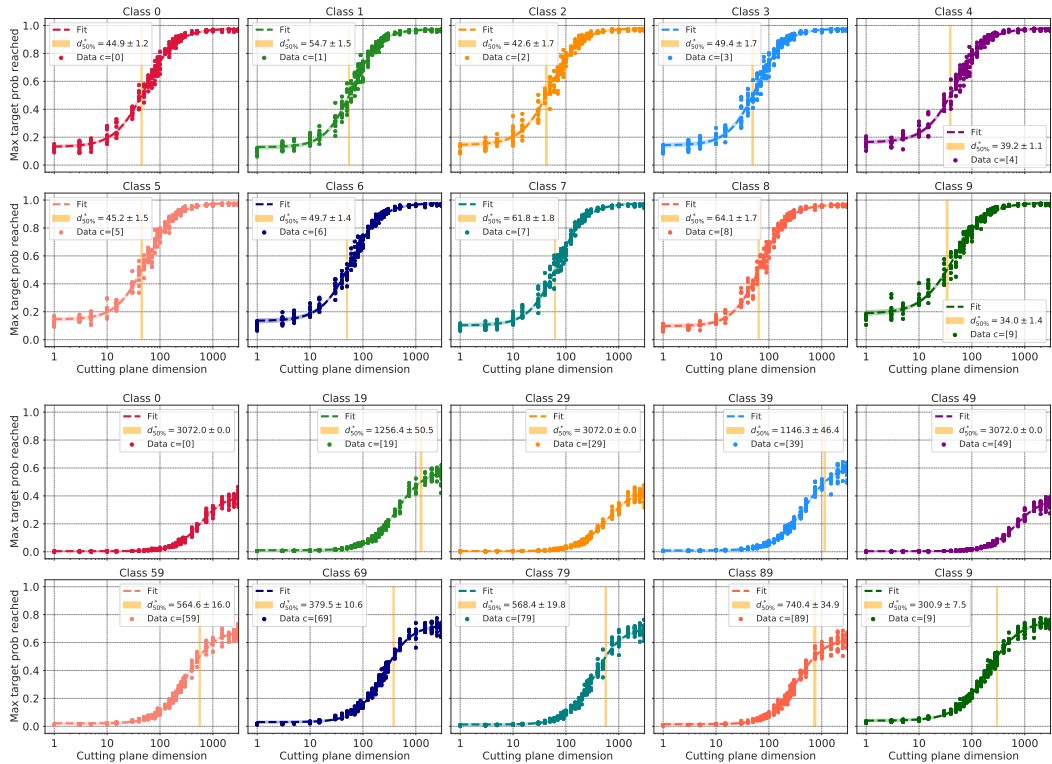

Figure 12: Maximum probability reached on cutting planes of different dimensions for all 10 target classes of CIFAR-10 (top row) and CIFAR-100 (bottom row) for a ResNet20v1 trained to 100% training accuracy on *randomly permuted* class labels. The $d_{50\%}^*$ is consistently higher and therefore the dimension of the high confidence manifolds is lower than for semantically meaningful labels (Figure 2), suggesting geometrically a very different function being learned.

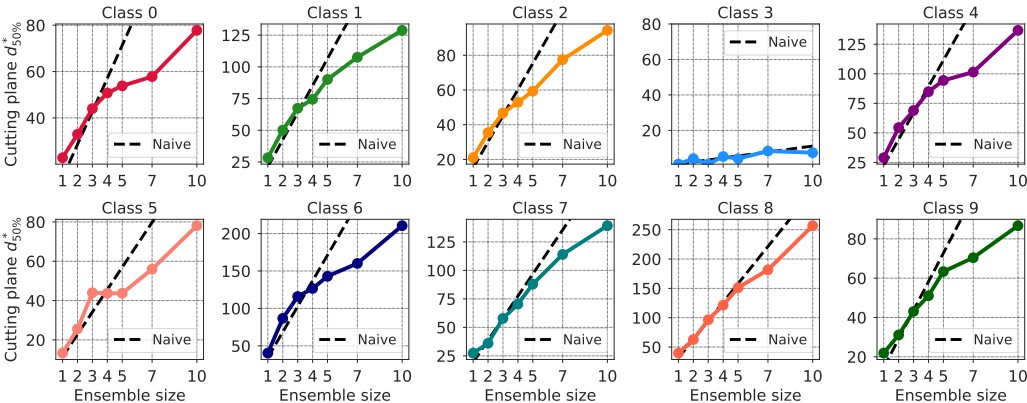

Figure 13: The effect of model ensembling on the dimension $d_{50\%}^*$ needed to reach $50\%$ accuracy of all 10 CIFAR-10 classes. The results shown are for ResNet20v1 trained for 50 epochs each. Universally across all classes, the larger the ensemble, the higher the $d_{50\%}^*$ and therefore the lower the high confidence manifold dimension. A naive model of addition of codimensions between models is overlayed, showing a surprisingly good fit for small ensembles.

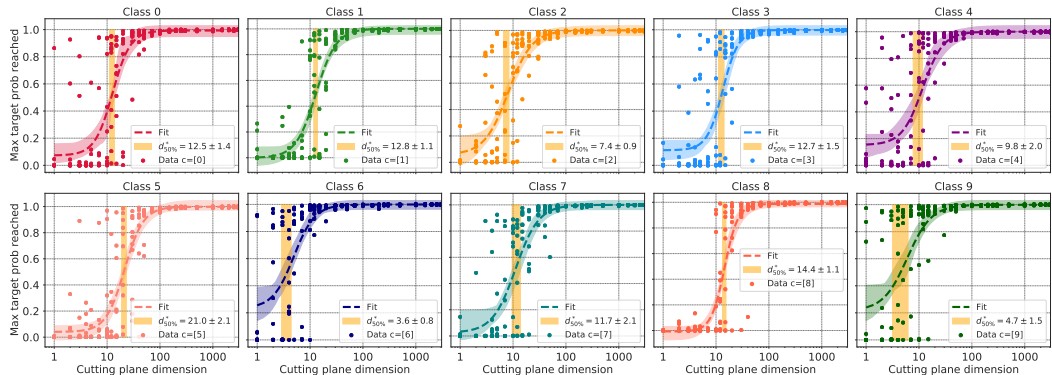

Figure 14: Maximum probability of single classes of CIFAR-10 reached on cutting planes of dimension $d$. The figure shows the dependence of the probability of a single class of CIFAR-10 (y-axes) reached on random cutting hyperplanes of different dimensions (x-axes). The results shown are for a well-trained ($> 76\%$ test accuracy) SimpleCNN on CIFAR-10. Each dimension $d$ is repeated $10\times$ with random planes and offsets.

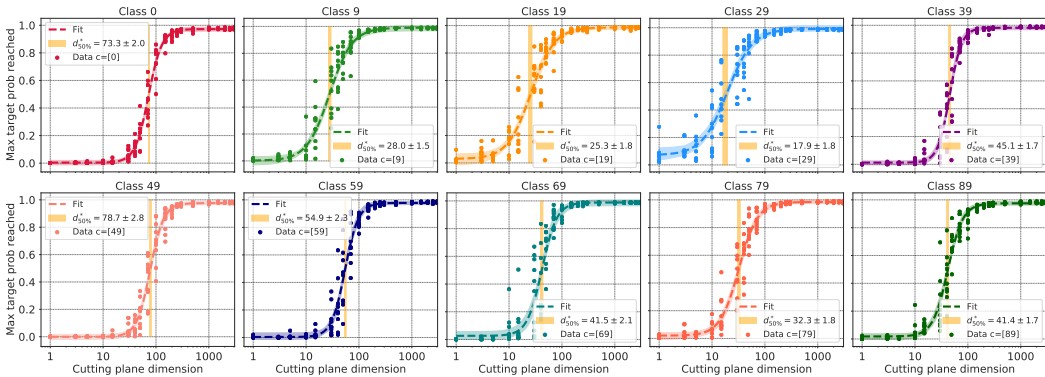

Figure 15: Maximum probability of selected single classes of CIFAR-100 reached on cutting planes of dimension $d$. The figure shows the dependence of the probability of a single class of CIFAR-100 (y-axes) reached on random cutting hyperplanes of different dimensions (x-axes). The results shown are for a well-trained ($> 67\%$ test accuracy) ResNet20v1 on CIFAR-100. Each dimension $d$ is repeated $10\times$ with random planes and offsets.

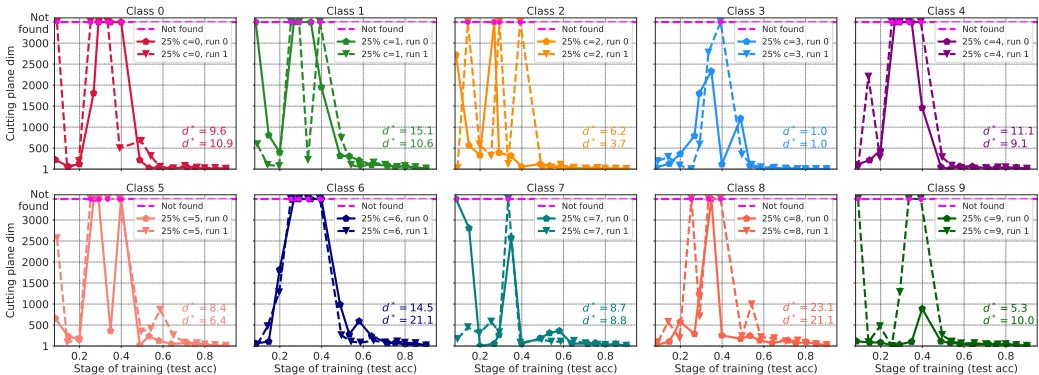

Figure 16: The cutting plane dimension needed to reach $25\%$ probability for the 10 classes of CIFAR-10 as a function of training stage for a ResNet20v1, averaged over two initializations and runs.

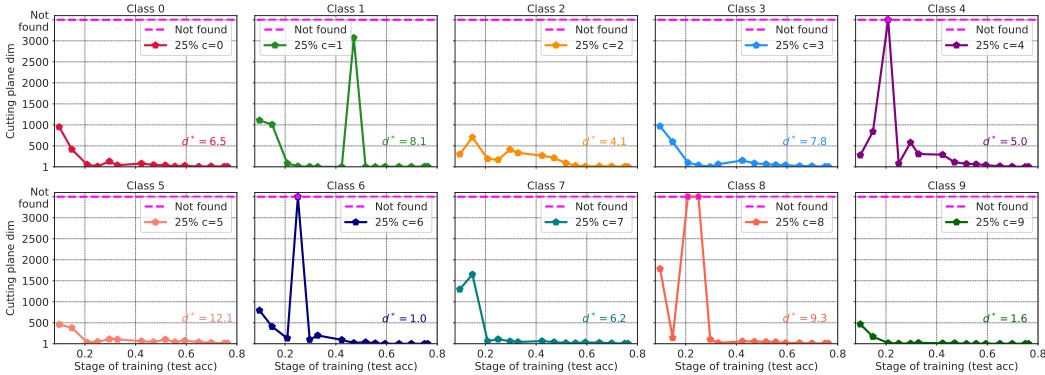

Figure 17: The cutting plane dimension needed to reach $25\%$ probability for the 10 classes of CIFAR-10 as a function of training stage for a SimpleCNN.

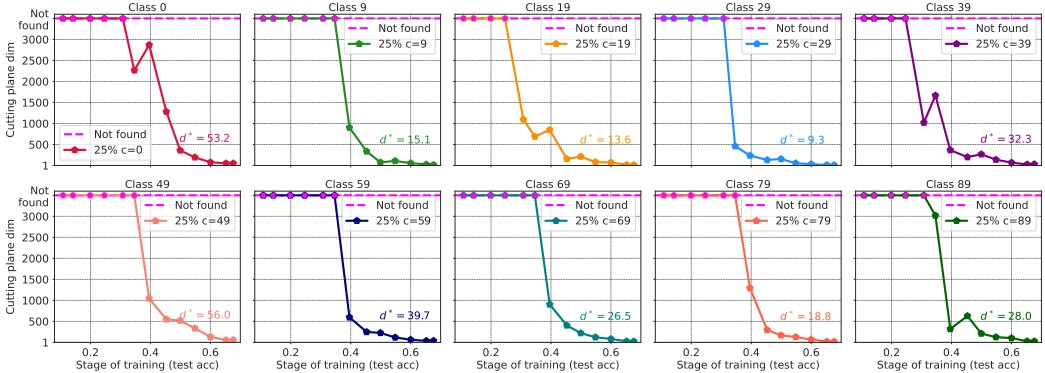

Figure 18: The cutting plane dimension needed to reach $25\%$ probability for 10 randomly selected classes of CIFAR-100 as a function of training stage for a fully trained ResNet20v1.

