# OpenReview forum: "Slice, Dice, and Optimize: Measuring the Dimension of Neural Network Class Manifolds"
_ICLR.cc/2021/Conference — Reject_

### Official Review · AnonReviewer2 · 2020-10-28
**Slice, Dice, and Optimize: Measuring the Dimension of Neural Network Class Manifolds**

**Rating:** 5
**Confidence:** 4

**Review:**

The authors propose a cutting plane method, inspired by intersection theory from algebraic geometry, to analyse the properties of neural networks. Specifically, the method allows to estimate the dimensionality of the class manifolds induced by the networks. An empirical analysis shows that the estimated dimensionality correlates with the generalisation performance and the robustness of neural networks, thus opening the door to potentially new perspectives on deep learning theory.

The paper is well structured and clearly written. Also, the authors are planning to release the code to reproduce their experiments. Last but not least in terms of importance, the paper provides an original and novel method for the analysis of the properties of neural networks. In fact, while previous works have used a similar strategy to estimate the intrinsic dimensionality of the loss landscape in the weight space (see cited works in the paper), this work focuses on the analysis of neural networks in the input space.

In general, there are no major issues with the paper. However, there are some points that need to be discussed, which can be helpful (i) to identify more precisely the conditions about the validity of their results and (ii) to relate with other existing work on the analysis of deep learning using spline theory. Please, see below for more detailed comments and also suggestions to increase the quality of the paper.

Based on these considerations, I recommend for the acceptance of the paper with an initial score of 6. I'm willing to considerably increase the score and award the authors, if they can address my questions.

DETAILED COMMENTS

Please, let me make two simple pedagogical examples to analyse the behaviour of the proposed method and to possibly seed further thought.

FIRST EXAMPLE

Consider a two-dimensional real space, where the class-manifold is a line. Then, generate a second line by randomly sampling its intercept and slope coefficient and refer to them as the line parameters. Now if you consider the parameter space of this second line, you have two regions, one of zero measure, which contains all the cases where the two lines are parallel, and the remaining one, which contains all the intersecting cases. This means that the two lines are almost always intersecting each other. Consequently, the estimated dimension of the class-manifold is correct.

SECOND EXAMPLE

Consider the same example as before, but now the class-manifold is a parabola. Similarly to the previous example, there are two regions, namely the ones defined by the intersecting and the non-intersecting cases between the parabola and the randomly generated line, but differently from the previous one, both regions have non-zero measure. Therefore, we may end up to generate lines that do not intersect the class manifold. This would result in considering a higher dimensional object (in this case a plane) to guarantee the intersection with the parabola. Consequently, we would underestimate the dimension of the class manifold.

This phenomenon can be even more pronounced when moving to higher dimensions. Therefore, I agree with the authors that the whole analysis is exact when considering hyperplanes. But what are its limitations when moving to the nonlinear regime? How can we guarantee that the estimated dimension is accurate? It seems that the proposed method provides a lower bound on the true dimensionality of the class manifolds. Is that correct? If so, when can this bound be tight?

Also, there is some recent line of work trying to analyse the behaviour of deep leaning in terms of decision boundaries and their curvatures from the perspective of spline theory [1-2]. Could you please discuss this and add the explanation in the paper?


SUGGESTIONS TO INCREASE THE QUALITY OF THE PAPER

I proceed following the order of the sections.

Section 2.2. Is it possible to provide the full details of the algorithm to estimate the cutting plane dimension, like an algorithmic table? Also, where is Equation 6 (in the appendix)? What are the mean and covariance parameters?

Section 3.1. Can you be more precise when you use the terms 'highly likely' and 'generically' and discuss what happens in the nonlinear regime?

Section 3.1. D-(d_A+d_B) should be 2D-(d_A+d_B)?

Section 3.1. Can you rephrase the sentence "For the subspaces A and B intersecting transversally...satisfying the upper bound and therefore leading to Equation 2" and make it more clear? Specifically, which upper bound and how does this lead to Equation 2?

Section 3.2. Can you consider to remove it, as the purpose is not clear and it does not seem to introduce any additional information?

Section 3.3. Can you make an example to explain the difference between dimension and effective dimension?

Section 4.3. Is there any concrete insight on the analysis of the class-boundary and multi-way class-boundary manifolds? I would appreciate to see more discussion on that.

Section 4.6. Is there any specific reason why you chose to show only class 0 and 1 in Figure 8? Can you provide the figures for the other classes as well, maybe in the appendix?

Section 4.7. Similarly to Figure 8, can you add the other cases for Figure 9? Always for this subsection, which initialisation did you use in the experiments? This is an important information that could be of interest for those studying initialisation strategies for deep learning?

Section 4.8. Do you have any experiment with ensemble of classifiers with different architectures? If so, do the same findings hold? May it be possible that you are underestimating the dimension of the class manifold in the set of experiments shown in the paper?

Section 4.8. Can you provide a plot of the generalisation performance versus the ensemble size? Or better correlating the cutting plane dimension with the generalisation performance for the different ensemble size?

[1] Balestriero and Baraniuk. A Spline Theory of Deep Learning. ICML 2018
[2] Balestriero et al. The Geometry of Deep Networks: Power Diagram Subdivision. NeurIPS 2019

#########################

UPDATE

The discussion phase has highlighted several major issues:
1. There has been a significant conceptual shift in the problem definition (i.e. from estimating the intrinsic dimensionality of the class manifold to quantifying its geometrical properties).
2. I'm not convinced about the validity of some arguments/statements used by the authors to support point 1. For example, the statement "The intrinsic dimensionality of class manifolds is absolutely the full dimension of ambient input space, but this is a completely uninteresting observation" is not fully supported and I'm not even sure that is true.
3. Furthermore, the paper is still in its original form. It has been difficult to keep track about the modifications that the authors should do.

To conclude, the article is not ready for publication, yet and therefore recommend for its rejection.

I encourage the authors to further investigate the topic and carefully consider whether the statements provided in the discussion phase are true.

---

> ### Author Response · Authors · 2020-11-19
> **A response to Reviewer 2 (part 1/3)**
>
> Thank you very much for your very detailed and insightful review! It actually motivated us to strengthen our argument, thank you! We’re happy that you think that our method could open the door to potentially new perspectives on deep learning theory, that you find our paper well structured and clearly written, and that you find our method original and novel. We hope that you will champion our paper!
>
> ## Code is public already:
> We would like to clarify that our code is already published (anonymously so far, of course) and that you can run the dimensionality estimates for single-class manifolds for all 10 classes of CIFAR-10 with ResNet20v1 within 5 minutes on a free Google Colab GPU: https://github.com/anonymous-code-submission/ICLR2021_paper333_anonymous_code_submission , replicating the main part of our paper!
>
> ##  Connection to spline theory
> Thank you very much for recommending [1] and [2], both of them are really good papers that we will refer to in the next iteration of our paper. If we understand [1] correctly, the local class-decision-boundary neighborhood of a MASO DN around a particular input is plane-like, and the full DNN can be well approximated using many such local approximations? This would be quite in line with the way we imagined the class manifolds would look like. Our method is sensitive to the global structure of the level sets of p[class] > threshold, which makes it quite distinct from [1] and [2], but it is very likely that the structure described in [1] and [2] locally builds up what we observe as the class manifold we are cutting with our planes. Both [1] and [2] are great papers and will likely be very relevant to our research going further. We will definitely refer to them in this paper and highlight the connection.
>
> [1] Balestriero and Baraniuk. A Spline Theory of Deep Learning. ICML 2018
>
> [2] Balestriero et al. The Geometry of Deep Networks: Power Diagram Subdivision. NeurIPS 2019
>
> ## Hitting non-planar manifolds
>
> Thank you very much for having such a detailed look at our paper and idea! As you confirm, the intersection of two randomly oriented affine subspaces of dimensions $d_A$ and $d_B$ generically exists provided $d_A + d_B \ge D$ where $D$ is the dimension of the ambient space.
>
> You came up with a great example, however, where this simple calculus is not exact: a 1 D parabolic line y = x^2 in 2D intersecting with a line y = ax + b for some (a,b). As you correctly state, the set of {(a,b)} such that there is no intersection is of non-zero measure, and so is the set of intersecting cuts. That means that were we to use our method here, for cutting planes of d=1, some would not find an intersection, while some would. For d=2, all of them would.
>
> First, we would like to mention that this makes hitting the manifold **harder**, leading to a higher d* for the cutting plane, and therefore to a lower estimate of the dimension of the class manifolds. One of the main surprising observations we consistently make in this paper is that the class manifolds are very high dimensional. Therefore if what we measure is actually an underestimate of the dimension and that already is very high, the true dimension is even higher! This actually further strengthens our empirical conclusions. Thank you for pointing this out.
>
> However, returning to the specific example of the parabola,  if we plot the full curve p(d) which is the probability of intersection p as a function of dimension of the cutting plane, over random choice of cutting plane for a fixed parabola, we would obtain in this example the curve p(0) = 0,   0 < p(1) < 1, and p(1) = 1.  Thus the curve would transition from 0 to 1 at the cutting plane dimension 1, which is the correct codimension of the parabola, which is 1. So in this particular example, our method can be viewed as a probabilistic method to estimate the correct co-dimension.
>
> One argument we can offer is that we are in fact not looking at sharp transitions in practice. The definition of dimension we work is effective or approximate  because we are not sure what the exact properties of the class manifold we are hitting are. In fact it is certainly full dimension in the strict sense of the word, but its extent in many of those dimensions is very small, effectively leading it to behave as lower than ambient dimensional in our experiments.
>
> ^^^^^^^^^^^^^^ CONTINUES IN THE NEXT BOX ^^^^^^^^^^^^^^

---

> ### Author Response · Authors · 2020-11-19
> **A response to Reviewer 2 (part 2/3)**
>
> However, we still want to address this question on its own. I believe that the crucial effect here is that the parabola does not extend between +- infinity in all directions. As such, we could model the same kind of phenomenon by having axis-aligned affine subspaces of dimension n, which are modified such that in each axis they only extend between 0 and +infinity, rather than -infinity and +infinity. Now the cutting plane needs to hit the manifold in all of its D-n “short” directions & at the same time have all the other axes >0, making the optimization harder. The condition on the expected dimension at which the intersection happens will therefore be  $d_A + d_B \ge D + K(d_A,d_B,D) $, where $K(d_A,d_B,D) > 0$ is a correction term.
>
> While we might be able to get it analytically, we instead ran a numerical simulation exactly as we did in Figure 4, but this time for target manifolds where we a certain number of the “long” dimensions to (0,infty) instead of (-infty,infty).  We then used the same setup as in Figure 4 and looked at what dimension we extract as we increase the number of non-full-range axes from 0 upwards.
>
> In a D=1000 dimensional space we made a n=900 dimensional target manifold in the form of an affine subspace. We ran 3 experiments with different numbers of axes out of the 900 that were restricted to (0,infinity): 0, 100 and 200. The results are shown here:
>
> https://github.com/anonymous-code-submission/ICLR2021_paper333_anonymous_code_submission/blob/master/toy_example_D1000_n900_halfdims0_reps3_id4527165.png
>
> https://github.com/anonymous-code-submission/ICLR2021_paper333_anonymous_code_submission/blob/master/toy_example_D1000_n900_halfdims100_reps3_id7713912.png
>
> https://github.com/anonymous-code-submission/ICLR2021_paper333_anonymous_code_submission/blob/master/toy_example_D1000_n900_halfdims200_reps3_id3544116.png
>
> For the 0 case, we estimated dimension n* = 897.4 (ground truth n=900). In the case of 100 out of the 900 axes being truncated to (0,infinity), we estimated n* = 896.5, which, within error, doesn’t seem to deviate from the ground truth of 900. When we shorted 200 out of the 900 axes, we got 848.3. There, the fact that the manifold is now harder to hit showed, as expected, as an underestimate of the intrinsic manifold dimension. In this particular case, we choose the cutting planes to start distance 1 from the origin, therefore putting the starting point near the places where the truncated dimensions of the manifold end. In reality, the starting point of the cutting plane would likely lie generically somewhere in the middle of such a truncated manifold, and the dimension underestimate would have been even better.
>
> The specific corrective factor $K(d_A,d_B,D)$ by which our estimate differs from the ground truth doesn’t seem to be very prominent in the experiment we ran and also would make our manifold dimension estimates lower bounds that are relatively close to their true values. Again, the fact that our lower bounds are indeed so high further strengthens our empirical conclusion that the manifold dimensions are very high.
>
> Thank you for bringing this up! We will definitely make sure these limitations are clearly stated in the paper.
>
> ## Finer points
> Thank you for the many suggestions you made to improve our paper. In terms of types, we will definitely correct those. We will also move the equation to the main text from the appendix.
>
> In Section 4.3, in terms of the interpretation of the interplay between the class and multi-class boundaries, we are still working on building a good geometric model. This paper is the first step in achieving that -- it gives us a tool and empirical results that we can build the theoretical model on.
>
> In Section 4.6 we show classes 0 and 1 only to save space. The trends we observe in other classes are equivalent and we have the data. We will add them to the appendix.
>
> In Figure 8 we only ran classes 0 and 1 because the experiments were very computationally expensive. Each dot is an average over many experiments. We will try to add more classes, but given the very weak class effect in all other experiments we performed, we do not expect any deviations from the trend here.
>
> In Section 4.8 the effect of combining different architectures in an ensemble will likely result in a very similar trend. If e.g. the c=0 manifold for architecture 1 has short directions {u_i} and architecture 2 has {v_i}, if those sets are independent their union will have the sum of their dimensions, leading to the same linear effect as observed in Figure 10. The experimental validation might be beyond the scope of this work, but it would likely behave in a very similar manner.
>
> In Section 4.8 we will add the generalization accuracy vs the dimension for the ensemble size. We have the data and we will make a new figure using them. The bigger the ensemble, the better the test accuracy, therefore we will see a correlation between the cutting plane d* and test accuracy.

---

> ### Author Response · Authors · 2020-11-19
> **A response to Reviewer 2 (part 3/3)**
>
> ##  Conclusion
> Thank you very much for a detailed and insightful review! Engaging with your questions actually made our thinking clearer and strengthened our empirical conclusions. We hope that you will champion our paper in the review process. If you have any more questions, let us know, we are happy to answer them.

---

> > ### Comment · AnonReviewer2 · 2020-11-22
> > **Reviewer Response**
> >
> > Thank you for the appreciation of my review and your answers.
> >
> > I went through the other reviews as well (in particular of reviewer 1 and 3) in order to better understand whether my judgement is correct.
> >
> > There seems to be a confusion in the reviews between estimating the intrinsic dimensionality of the data and estimating the dimensionality of the class manifold induced by the network. These two concepts are different as the first one is typically low (referred to the dimensionality of the data manifold, i.e. the manifold assumption), whereas the second one is generally high (as confirmed by reviewer 3 with the random input example).
> >
> > Now, it is clear that the estimator you propose is a lower bound on the dimensionality of the class manifold, but it's still not clear to me whether this bound is tight in practice. There are several factors, apart the ones that I was mentioning in the review, like the "curvature of the class manifold, its boundedness, its disconnectedness..." (as noted by reviewer 2) which could lead to a poor lower bound estimate. On one hand, it could be that the class manifold has TRUE dimensionality equal to the ambient space. If that is the case, then the analysis of the paper looses its significance, because the lower bound does not provide any useful information and therefore I have to agree with reviewer 3. But, on the other hand, if the TRUE dimensionality of the class manifold is lower than the dimensionality of the ambient space, your analysis provides several interesting insights, which are worth of being published and I'm willing to defend during the next phase. Can you please elaborate more on that and clarify this aspect?

---

> > > ### Author Response · Authors · 2020-11-23
> > > **A response^2 to Reviewer 2 with relevance to other reviewers (part 1/4)**
> > >
> > > Thank you for getting back to us so quickly.  We would also like to thank you as well as the rest of the reviewers for such an interesting and thought provoking review period. The process has led us to realize that several key pieces of phrasing in our paper were a bit ambiguous. We feel this ambiguity on our part has caused the review process to focus on technical questions that are besides the point of our paper. We would like to clarify our contribution and terminology in light of this very productive discussion.
> > >
> > > In particular, our use of the terms “dimension” and “co-dimension” as stand-ins for more subtle characteristics of the class manifold have caused confusion. We hope to clarify the perspective taken in our submission once-and-for-all.
> > >
> > >
> > > ## 1. We are focusing on learned class manifolds not data manifolds:
> > >
> > > The goal of this paper was never to explore the geometry of real data manifolds (i.e. the manifold of images that correspond to **real** cats).  Other work has explored this, and a general consensus in the field is that this manifold has low intrinsic dimensionality relative to the number of pixels.  We are instead focusing on the **anisotropic geometry** (not the intrinsic dimensionality) of class manifolds, or regions of input space that a trained deep network assigns a particular class label with high probability.
> > >
> > > ## 2. The intrinsic dimensionality of class manifolds is absolutely the full dimension of ambient input space, but this is a completely uninteresting observation and not at all related to the fundamental point of our work.
> > >
> > > If p(c|x) is the probability a given image x in pixel space is assigned class c by a neural network, the class manifold is simply the set of points $p(c|x)>1-\epsilon$ where $\epsilon$ is small but strictly nonzero.  Because this class manifold is defined by an inequality, as long as the probability map p(c|x) is continuous, which it generically is for neural networks used in practice, then at a point $x$, one can always move infinitesimally in all N directions where N is the number of pixels (times the number of channels) without leaving the class manifold.  Therefore the intrinsic dimension of the class manifold is equal to the ambient dimensionality of space.  However, this fact is completely uninteresting, as it depends only on the input-output map of the neural network being continuous. Thus the intrinsic dimension of class manifolds cannot co-vary and therefore cannot be diagnostic of many quantities of interest that practitioners care about - i.e.  1) architecture, 2) dataset, 3) the amount of training, 4) dataset size, 5) data augmentation, 6) label randomization, 7) robustness to noise and perturbations, and 8) ensemble size. While we discussed this point in our submission, we still used nomenclature related to intrinsic dimensionality in our submission. We intend to remove this terminology and clarify this point.
> > >
> > > ^^^^^^^ CONTINUES IN THE NEXT TEXT BOX ^^^^^^^^

---

> > > ### Author Response · Authors · 2020-11-23
> > > **A response^2 to Reviewer 2 with relevance to other reviewers (part 2/4)**
> > >
> > > ## 3.Third: the fundamental point of our work is to probe the much more interesting anisotropic geometry of class manifolds, not their intrinsic dimensionality, which we have already established is a completely uninteresting property.
> > >
> > > What is much more interesting is the **anisotropic geometry** of this class manifold rather than the intrinsic dimension.  Famous and well known prior work has already shown that this manifold can have a highly anisotropic geometry.  For example, starting from a correctly labelled image belonging to a high confidence region of a particular class manifold, if one moves in an adversarially chosen dimension, then one can leave this manifold after only moving a very short distance, hence the existence of adversarial examples. Conversely, in a random direction, one must move a much larger distance to leave the class manifold.
> > >
> > > Moreover, In our Figure 10, we show a 2D section of the input space for 3 models (and their ensembles). You can clearly see that class manifolds shown there (the colors indicate class probabilities >=50%) are much longer in some directions than others. That implies that among the 3072 directions of the class manifolds (CIFAR-10), the shorter directions couldn’t have been that rare, otherwise they would have never appeared on a 2D section like this.
> > >
> > >
> > > Thus class manifolds have a highly anisotropic geometry with many extended and unextended directions.  Probing this **geometric anisotropy** more carefully is therefore of fundamental interest to us and this is the main point of this paper.
> > >
> > > For example, even if the intrinsic dimension of all class manifolds is the full ambient dimension, this observation leaves open drastically different possibilities for the actual **geometry** of the class manifold.  For example, one extreme possibility is that the class manifold may be a union of full ambient dimensional but tiny epsilon balls surrounding a much lower intrinsic dimensional data manifold of images from that class. In this situation, the class manifold is simply a limited fattening of the data manifold to extend a small amount in all dimensions.  Conversely, the class manifold could extend far away from the data manifold in many different dimensions, assigning a high probability p(c|x) to images x far from the data manifold for that class.  Both situations have the same intrinsic dimension (again equal to the full ambient dimension) but reflect radically different geometries.
> > >
> > > Our approach to more fully probe the nontrivial anisotropic geometry of class manifolds rested on a cutting plane technique: namely starting from a random point $X$ (from the training set), optimizing within a hyperplane (affine subspace) of dimension $d$ to enter the class manifold, and recording the probability of entry $p(d)$ which is an entire curve that provides considerable information about the geometry of the manifold and how easy it is to hit it.
> > >
> > > We attempted to summarize this entire curve $p(d)$ with a single number, $d^*$ related to when it transitions from 0 to 1 as a function of $d$.   In  the very special motivating case in which the class manifold is a hyperplane (affine subspace) of intrinsic dimension K, this curve transitions sharply at a point equal to the co-dimension of  the hyperplane, namely $N-K$, and therefore we sloppily called $N-d^*$ the estimated dimension of the class manifold.  This name was a regrettable misnomer which has led to understandable confusion during the review process.  Indeed $d^*$ is a much more interesting property, as it probes the **anisotropic geometry** of the class manifold, and in particular, the number of **extended directions** it has.

---

> > > ### Author Response · Authors · 2020-11-23
> > > **A response^2 to Reviewer 2 with relevance to other reviewers (part 3/4)**
> > >
> > > Consider for example the simple case where the **data manifold** (not the class manifold) is a hyperplane of intrinsic dimension $K$ in $N$ dimensional pixel space.  But then consider an ideal neural network that creates a **class manifold** consisting of all image points x that are within a tiny distance $\epsilon$ of this data manifold in the full $N$ dimensional pixel space.  This class manifold has intrinsic dimension equal to the full ambient dimension $N$. However, its geometry is highly anisotropic: it is greatly extended and long in $K$ dimensions, and is greatly restricted and thin in $N-K$ directions.  If one were to employ our cutting plane technique to explore the geometry of this class manifold, one would find that the resultant curve $p(d)$ would now transition more smoothly from 0 to 1 approximately at a location $d^* =  N-K$.  The width of the transition region is directly related to the thickness of the thin directions; the transition is infinitely sharp only when the hyperplane is infinitely thin and extended. Thus the intrinsic dimension of the class manifold takes the completely uninteresting value of $N$, while the entire cutting plane curve $p(d)$ and the  $d^*$ value at which it transitions is **much more interesting**, as one can read off of it the actual number of extended dimensions in the class manifold, via $K \approx  N - d^*$.
> > >
> > > It is in this sense that our cutting plane method **was fundamentally designed to probe the anisotropic geometry, not the intrinsic dimensionality of class manifolds**, and in many situations, it approximately corresponds to measuring along how many independent directions the class manifold has significant extent.
> > >
> > > ## 4. Any attempt to evaluate the utility of our work based on how tight a bound our measure is on the intrinsic dimensionality of class manifolds misses the fundamental point of our work.
> > >
> > > Because of our misnomer of calling $d^*$ the co-dimension of the class manifold, and $N-d^*$ the effective dimension of the class manifold, this review process has gone down a conceptual rabbit hole of attempting to assess the value of our work based on how tight of a bound $N-d^*$ is on the intrinsic dimension of the class manifold.  Based on our description above, we now hope the reviewers will agree that this is a misguided question and is an inappropriate way to evaluate our work.  As discussed above, $d^*$ (and indeed the entire curve $p(d)$) is fundamentally designed to probe the much more interesting **anisotropic geometry** of the class manifold, and **not** the intrinsic dimensionality.  The example of the fat hyperplane class manifold should make this clear, where the intrinsic dimension is $N$ (and is uninteresting) but $d^*$ is $N-K$, and is much more interesting as it tells you roughly the number of directions in which the class manifold is thin.
> > >
> > > Thus we hope now Reviewer 2 appreciates that their following concern: “Now, it is clear that the estimator you propose is a lower bound on the dimensionality of the class manifold, but it's still not clear to me whether this bound is tight in practice,” might not be the right question to ask as long as they think about the linear algebraic definition of a dimension, rather than the effective dimension we are considering.  Indeed $N-d^*$ is intended to measure entirely different and more interesting geometric properties of the class manifold rather than its intrinsic dimension. **Therefore how close it comes to the intrinsic dimension is completely immaterial to the value of the measure.**  Furthermore we note that Reviewer 2 also asserts that “if the class manifold has TRUE dimensionality equal to the ambient space,” then “the analysis of the paper looses its significance, because the lower bound does not provide any useful information and therefore I have to agree with reviewer 3.”  **We hope both Rev. 2 and Rev. 3 now appreciate that this assertion is inappropriate because it is based on two fundamentally incorrect assumptions:** (1) the intrinsic dimension of the class manifold being equal to the ambient space is an interesting observation (it is not as it is simply tantamount to the statement that the neural-input output map is continuous); and (2) our methods and goals were fundamentally to lower bound this intrinsic dimension (they were not - our goals were to instead to probe the anisotropic geometry of class manifolds, not their uninteresting intrinsic dimension).

---

> > > ### Author Response · Authors · 2020-11-23
> > > **A response^2 to Reviewer 2 with relevance to other reviewers (part 4/4)**
> > >
> > > Our cutting plane method does indeed probe the anisotropic geometry of class manifolds by providing information on roughly how many dimensions are extended. Surprisingly, we found as many as almost 3000 directions are quite extended.  Based on the consensus view that data manifolds are low dimensional, at least presumably lower than 3000, this provides quantitative evidence for the assertion that class manifolds do not hug data manifolds (in the one extreme case posited above), and the scenario is much closer to the other extreme example, where class manifolds extend in many more directions than data manifolds do, and neural networks overconfidently assign high probability class labels to images far from the veridical data manifold.
> > >
> > > And most interestingly our measure correlates with and can therefore diagnose many aspects of neural networks that people care about in practice, including: 1) architecture, 2) dataset, 3) the amount of training, 4) dataset size, 5) data augmentation, 6) label randomization, 7) robustness to noise and perturbations, and 8) ensemble size.  See e.g. training set size (fig. 7), robustness (fig. 8), and training time (fig. 9).  The intrinsic dimension correlates with none of these quantities. Therefore we believe our cutting plane method is of demonstrable empirical utility in understanding the input-output maps derived by deep learning.
> > >
> > > ## 5. Our effective manifold dimension $d^*$ is closely related to the *stable dimension* derived using the concept of Gaussian width in high-dimensional statistics
> > >
> > > Our concept of the effective dimension, namely ($N-d^*$) is quite closely related to the well established concept of the **stable dimension** as described in the book **High-Dimensional Probability: An Introduction with Applications in Data Science** by Roman Vershynin [1], where it is closely related to the Gaussian width of a subset of $\mathbr{R}^{D}$. There the author notes that:
> > >
> > > “The linear algebraic dimension is unstable: it can significantly change (usually upwards) under a small perturbation of $T$ [the subset of points]. A more stable version of dimension can be defined based on the concept of Gaussian width.” In the defintion of the stable dimension, we look at the width of the set of points = our class manifold averaged over all directions. This is very closely related to the notion of “short” and “long” directions that we use conceptually in our paper. Therefore our notion of effective dimension has some related precedent with theoretical justification, though is not identical to it.
> > >
> > > [1] Roman Vershynin. High-Dimensional Probability: An Introduction with Applications in Data Science. Cambridge University Press, 2018.
> > >
> > > ## Finally:
> > > We again thank the reviewers for their careful reviews, and we hope this careful response clarifies the misunderstandings we may have inadvertently generated during the review process by initially engaging on issues of dimensionality bounds and their tightness, which we don’t believe is fundamentally  relevant to our work.  We will add these clarifications to the camera ready version of the paper, and we hope we have clarified our positions.  We are happy to engage further in case we have not.

---

### Official Review · AnonReviewer1 · 2020-10-30
**Dimensionality estimates are too high?**

**Rating:** 4
**Confidence:** 4

**Review:**

This paper proposes an empirical method for estimating the dimensionality of a class manifold, defined here as a collection of points for which the last (softmax) layer of a neural network maps them to a membership probability vector associated with a specific class. Their approach involves the generation of a randomly oriented 'cutting plane' of dimension $d$, passing through a randomly generated source point. The authors note that if the sum of the dimensions of the class manifold and the cutting plane exceeds the full spatial dimension, the chance of an intersection of the two is high.  Conversely, if the sum falls short of the full dimension, the chance of an intersection is very low.

Using a gradient descent technique starting at the source point, a location within the cutting plane is sought that minimizes the cross entropy loss between it and a target class membership vector representing the class manifold. A low minimum loss would indicate a likely intersection ($d$ too high), whereas a high loss would indicate a likely miss ($d$ too low). Although the dimension of the class manifold is in general unknown, the process is iterated for many choices of the initial cutting plane, and many choices of the cutting plane dimension $d$. The value of $d$ achieving the median loss value is chosen as the estimate of dimensionality.

In their experimental validation of the approach, the authors examine the effects on the estimated manifold dimensionality due to various factors, including data noise, label noise, training set size. Interestingly, their method also allows them to produce an estimate of the class boundary dimensionality by specifying the average of two class one-hot vectors as the target probability vector.

Pros:
-----

1) This is an interesting approach to the problem of dimensional modeling.  Estimation of dimensionality using cutting planes, without an explicit parameterization of the subspace, is an attractive idea that (if performed efficiently and reliably) could be particularly impactful. A strong point of the model is that it considers as its 'class manifold' the regions of deep latent space that have sufficient probabilities of being in one or more classes of interest. The method thus supports assessments of dimensionality in border regions in an elegant way.

2) The optimization procedure proposed does seem practical enough - each optimization run is efficient, and the number of runs can be tailored to an execution budget.

3) The paper is generally well organized and presented. The descriptions are clear and accessible.

Cons:
-----

1) As acknowledged by the authors in the caption of Fig 2, the dimensional estimates seem much higher than the typical estimates of intrinsic dimensionality as determined by local estimators (e.g. LID / Levina & Bickel, etc). This discrepancy could be due to a number of factors that are not taken into account: curvature of the class manifold, its boundedness, its disconnectedness, etc. All these factors could cause the gradient descent to terminate at high cross-entropy loss values, which would drive the estimate of dimensionality too high (even approaching the representational dimension of the latent space?).

2) Following from 1), some of the conclusions reached from the experimental analysis are not fully convincing. For example, in 4.5 an inverse relationship is reported between the training set size and the 'effective' dimension. However, non-uniformity of distribution within the manifold could lead to configurations that trap solutions at unrealistically-high values of $d$.  In 4.6, adding Gaussian noise to each pixel is a full-dimensional transformation that is known to strongly bias the local intrinsic dimensionality upward, to unrealistically high values.

3) Again following from 1), the authors have not situated their work with respect to the recent literature on the use of intrinsic dimensional estimation in deep learning settings. For example, local intrinsic dimensionality has been proposed as a characterization of learning performance (Ma et al, ICML 2018), adversarial perturbation (Ma et al, ICLR 2018, Amsaleg et al, WIFS 2017), and in GAN-based image infilling (Li et al, IJCAI 2019). How does their estimator compare in practice to other estimators already in use?

Other comments / questions:
---------------------------

1) The paper should be more self-contained in places. For example, Equation 6 is referred to in the main paper, but appears only in the appendix.

2) Like distances distributions themselves, loss functions may exhibit a bias due to the local intrinsic dimensionality. Discuss?

---

> ### Author Response · Authors · 2020-11-19
> **A response to Reviewer 1 (part 1/2)**
>
> Thank you for your detailed review! We’re very happy that you find our approach to dimensionality estimation interesting, the idea attractive and that you appreciate the elegant way our method enables us to study the border regions between any set of classes. We also appreciate that you like the efficiency of our method and the presentation of our paper.
>
> In the following we will address the points you raised in your review and we will hopefully be able to clarify the points of confusion.
>
> ## Comparison to local dimensionality estimates:
>
> You are correct in saying that the dimensions that we estimate for single class manifolds are relatively high. When we point this out in the paper, we did not mean to compare to any previous values but rather to the **naive expectation that out of the 32*32*3 = 3072 ambient space dimension CIFAR-10 the dimension could be anywhere between 1 and 3072**, while we typically observe a number that’s much closer to 3072 rather than 1. We will be clearer that the surprise we show is not meant to suggest a difference to an existing method but rather to the naive, and not well grounded expectation that we had and the readers might share before reading our work.
>
> In your review you suggest that a number of factors such as curvature, boundedness, its disconnectedness “could cause the gradient descent to terminate at high cross-entropy loss values”. We agree with this sentiment. However, note that in many ways our estimate of the class manifold dimension is a lower bound. In particular, if **hitting the class manifold becomes *harder*, one needs a higher cutting plane dimension d*. This implies a measurement of  the class manifold dimension, as D - d*, that is lower than “true” manifold dimension**. Therefore the arguments you provide even strengthen our empirical observation that the manifolds are have a high dimension -- if, as you say, “these factors could cause the gradient descent to terminate at high cross-entropy loss values”, we would consistently be **underestimating the dimensionality of our manifolds**, not overestimating them, thereby strengthening our empirical conclusions.
>
> In terms of comparison to local dimensionality estimators, thank you very much for pointing them out. We are very happy to include a discussion comparing the LID method to our proposal here. We looked at [1] in detail and would like to point out that LID often suffers from dimensionality underestimation due to finite sampling. The method is also very different from ours in, both in terms of its locality versus our globality, and the actual estimation algorithm. While both have their pros and cons, we argue that our estimate is a better representation of the properties of the full function the NN has learned from the data, rather than a small neighborhood around a particular datapoint.
>
> Would you please give us a set of papers that make estimates of the dimension of learned class manifolds that we could compare to? The paper you suggested [1] presents a method, however, we were not able to locate any specific empirical results that would relate to our results there.
>
>
>
> ## Training set size vs dimension
>
> In your review you mention that “inverse relationship is reported between the training set size and the 'effective' dimension”. We would like to point out that you might have misread the figures. We report an inverse relationship between the training set size and d* = the cutting plane dimension needed to hit a class manifold which is the estimated manifold **CODIMENSION**. Therefore we show that the **manifold dimension goes down with training size**.
>
> Similar to the argument you make in the previous point, when you say  “non-uniformity of distribution within the manifold could lead to configurations that trap solutions at unrealistically-high values” this would only make the manifold harder to hit, leading to a higher cutting plane d*, and therefore to a lower estimate of the dimension of the manifolds. If that’s indeed what happens, and we **still** observe the cutting plane d* to go down with training set size, it only makes our observation stronger -- the real d*, if you are right, are even smaller, and therefore the dimension of the manifolds even higher.

---

> ### Author Response · Authors · 2020-11-19
> **A response to Review 1 (part 2/2)**
>
> ## Gaussian noise addition
> Thank you for pointing out that Gaussian noise increases the dimension that gets estimated by local methods. It is very likely that our method is **very resistant to such local perturbations** as we aim at the full, large-scale manifold and its properties rather than the local neighborhood of a particular point that might indeed get significantly affected. This is in fact a significant advantage of our method.
>
> We would like to clarify that in Figure 8 where we discuss the Gaussian perturbation, **we always keep the Gaussian perturbation magnitude constant, as is standard in common corruption benchmarks**. What varies are the different models that we test, which are all trained on clean, unperturbed data. We observe the better a model is at resisting the Gaussian noise perturbation in terms of generalization, the higher the dimension of its class manifolds. We will try to clarify this further in the text to avoid this confusion.
>
> What could also be helpful is Figure 11 in the Appendix in which we study the effect restricting our cutting planes to only a few pixels has on the dimension estimates we obtain, showing that unless only a very small number of pixels is varied, our estimates quickly converge to the correct value.
>
> In terms of stability to Gaussian perturbations, our notion of dimensionality is more similar to the **stable dimension** as defined in *High-Dimensional Probability: An Introduction with Applications in Data Science* [A] using the concept of a Gaussian width of a subset of $\mathbb{R}^D$ rather than the brittle linear algebraic dimension that you might have been imaging. We will make this clearer in the text.
>
> [A] Roman Vershynin. High-Dimensional Probability: An Introduction with Applications in Data Science. Cambridge University Press, 2018.
>
> ## Comparison to local estimators
>
> Thank you very much for pointing out the papers on local estimators. We will mention them in our paper and relate our contribution to them. Would you mind giving us more concrete references?
>
> [1] has a very different method, and does not provide actual estimates for class manifolds for the network and datasets we looked at. [2] uses a very similar method to [1] based on counting points in the neighborhood of another and seeing what their statistics looks like as a function distance. We could not identify which (Ma et al, ICLR 2018) you meant, could you please be more specific? [3] again uses local estimates based on [1] and which depend on the statistics of nearest neighbors vs their distance. This is very different from what we do both in terms of the locality and the actual algorithm. [4] uses the estimator from [1] again.
>
> Thank you for recommending these papers for us to compare to. We will make sure to refer to them in our work. All of them, however, are based on local, neighborhood distance statistics estimates, which are very different from our global method. Our contribution is therefore not affected.
>
> [1] Elizaveta Levina, Peter J. Bickel. Maximum Likelihood Estimation of Intrinsic Dimension
>
> [2] Characterizing Adversarial Subspaces Using Local Intrinsic Dimensionality. https://arxiv.org/abs/1801.02613
>
> [3] The Vulnerability of Learning to Adversarial Perturbation Increases with Intrinsic Dimensionality by Amsaleg et al. (https://www.nii.ac.jp/TechReports/public_html/16-005E.pdf)
>
> [4] Generative Image Inpainting with Submanifold Alignment https://arxiv.org/abs/1908.00211

---

### Official Review · AnonReviewer3 · 2020-11-01
**Interesting and solid work, but the fundamental assumption seems problematic**

**Rating:** 4
**Confidence:** 4

**Review:**

This paper proposes to understand the behavior of deep networks for classification tasks by studying the dimensionality of the "class manifolds", i.e., regions in the data space that are mapped to the same one-hot output. To measure such dimensionality, the paper proposes a method that is based on intersecting the class manifold with a random affine subspace of varying dimension. The idea is that when there is a intersection then the dimension of the random affine subspace is roughly the codimension of the class manifold. The paper then studies how different factors in data, architecture, training, etc., affects such dimensionality.

Strength:

The development of the paper is solid in the sense that it studies the effect of a wide range of design choices (see the list 1) - 9) in paper abstract).

Weakness:

The whole paper is based on the assumption that each "class manifold" is a low-dimensional manifold. However, the paper did not provide a justification for this assumption nor do I think it is a valid assumption.

The manifold assumption is a fundamental assumption for machine learning and data science, and that assumption is made for *data*, rather than *classes learned by neural networks*. One intuitive justification for that assumption in the case of data is that if I take a data point (say an image) and I do a perturbation, then if the perturbation is in a direction that is "meaningful", say by a translation, rotation and distortion, then the class label for that data point remains the same, but if you go towards another direction then likely the image is no longer meaningful and the class label changes. However, this same line of argument does not seem hold for the class manifolds learned by neural networks: if I consider a random input to a network then because the decision boundary is piecewise linear, it is with high probability that you can go towards all directions and maintain the class label.

If the low-dimensionality assumption is not valid, then the premise of the entire paper becomes problematic: the intuition given in Fig. 1 is no longer valid, and the theory in Sec. 3 is no longer meaningful.

Even if the low-dimensionality assumption is true to some degree, the proposed dimension estimation is still very much problematic: both the intuition in Fig. 1 and the theory in Sec. 3 are based on assuming that such low-dimensional manifold is (close to being) linear. But the ability of deep networks for performing complicated nonlinear mapping, which is the key to its great success, likely makes such low-dimensional manifolds to be highly nonlinear. Therefore, a discussion of how such nonlinearity affects the proposed dimension estimation is quite necessary but is missing.

Additional comments:

- How is X_0 in the cutting plane method generated? It is said in the paper that it is generated at random, so perhaps that means a i.i.d. Gaussian vector, but presumably the variance of the Gaussian distribution could have an impact on the result as it captures how far the affine subspace is from the origin.

- Sec. 3.1, which contains the main theoretical result of the paper, is presented in vague terms (e.g., what is "highly likely", under what statistical model?). Perhaps it is better to make it precise by writing the result as a theorem.


**Update after rebuttal**

I would like to thank the authors for the detailed rebuttal, but my feeling now is that the rebuttal is making it even more complicated and sometimes conflicting with itself. I believe the paper needs some careful rewriting and updates to clarify its points and assumptions.

Concretely, the paper is built upon the premise that each class manifold is a submanifold with dimension lower than that of the ambient space. I pointed out in my review that this premise may not hold at al, therefore the paper is fundamentally problematic. Then, R2 in one of his/her responses raise the same question, perhaps after reading my question. Then, I see a difference in response to R2 and my comments. For R2, the response is "The intrinsic dimensionality of class manifolds is absolutely the full dimension of ambient input space", which is effectively acknowledging that my critique is valid. However, the response to me is "This is very easily refuted by the ubiquitous and universal existence of adversarial examples". I don't really see why there is a discrepancy here. Besides, the argument that is used to refute my argument, namely existence of adversarial examples implies class manifolds are lower dimensional than the ambient space, is apparently wrong and can be easily refuted. By and large, the existence of adversarial examples only means that the decision regions are thin at every location, I can totally have a fine mesh of the data space that achieves this.

---

> ### Author Response · Authors · 2020-11-10
> **A response to Reviewer 3**
>
> Thank you for your detailed review. We feel that there is a **very serious misunderstanding** that we would like to rectify in this comment. Given that it forms your main objection to our paper, we would like to ask you to reconsider the score you gave us. We will also address your other comments.
>
> ### The low dimensionality of class manifolds is **not assumed**, it is **not needed**, and it is also **not observed**
> In your review you suggest that in our paper we
> 1) assume that the class manifolds are low dimensional and
> 2) that this assumption is needed for our method to work.
> **Neither of these statements is in fact correct.** Our empirical measurements show the exact opposite: the class manifolds are surprisingly high dimensional and our method is agnostic to this.
>
> #### Toy model with affine subspaces and known dimensions:
> Our method is agnostic to the dimension of the class manifold we are trying to hit. Let’s say we work in the idealized case where the target manifold is an affine subspace of dimension $n$ in a space of dimension $D \ge n$ (the case in Figure 4). The cutting plane dimension $d^*$ need to hit it reliably would be $d^* = D-n$ as stated in Eq. 2 and derived in detail in Section A.2. For example, if we’re in a D=1000-dimensional space and the manifold we’re hitting has n=900 dimensions (high dimensional manifold), we need the cutting plane of dimension at least d=100 (precisely the setup in Figure 4). If the manifold is relatively low-dimensional, e.g. n=50, we need d>950 to hit it. **Our cutting plane method is absolutely agnostic to whether the target manifold is low or high dimensional.**
>
> #### Our measurements show class manifolds to be very **high** dimensional
> In fact all the actual measurements we make show very high dimensions for the class manifolds. The very first results in Figure 2 include a line saying
> *"Indeed their dimensions are all in excess of 3000"*,
> out of the 3072 dimensions of the CIFAR-10 32x32x3 images. You can see that for cutting planes of e.g. d>72 you get p_reached > 50% for each of the classes / class manifolds. This is the case for all our results -- in all our results the class manifolds are very high dimensional, **exactly the opposite of what you are suggesting** in your comment.
>
> ### Nonlinear manifolds:
> Our theory in Section 3 is simpler with linear manifolds and that’s why we use them in our derivation. You are right that in practise the class manifolds are not just affine subspaces. However, the same dimensionality condition for intersection (as specified in Equation 2 for affine subspaces and in Equation 3 for generic manifolds) holds for generic manifolds as well. In algebraic geometry this is known as *dimension counting*. If the random cutting plane is in fact random, it does not preferentially align with the dimensions of the target manifold, and therefore generically the condition that codim(cut intersect manifold) = codim(cut) + codim(manifold) holds. This is a known result and that’s why we didn’t rederive it here (we have a reference there for it). Our theory is derived using affine subspaces, but the same results holds for generic nonlinear manifolds as well. It would definitely depend on their compactness as well, that's why we talk about effective dimensions, where this aspect is already absorbed into our dimension estimate.
>
> ### Sampling the starting image X0 from train set
> This is a great question, thank you. We experimented with a few ways of getting a starting point X0, including a Gaussian normalized to have the same L2 norm as the typical datapoint. However, **we actually use train set images of different than target classes** (to make sure we do not start at the finish line already) as the starting points, to stay as close to the manifold as possible and pick up the dimension there. You can verify this in our publicly available code at https://github.com/anonymous-code-submission/ICLR2021_paper333_anonymous_code_submission. We show this in Figure 11 but we will make sure this is clearer in the text. Thank you!

---

> > ### Comment · AnonReviewer3 · 2020-11-18
> > **Clarifications on my question**
> >
> > I would like to thank the authors for the detailed response.
> >
> > I don't think I have any misunderstanding as claimed by the response. Rather, I think the authors have a misinterpretation of my comment. Although, possibly I should be blamed for not making my comment crystal clear. So, please let me try again.
> >
> > The entire paper is built upon the assumption that each class lies in a *manifold*. In fact the paper aims to estimate the dimension, therefore it really means *sub-manifold* **whose dimension is lower than the ambient space dimension**. However, there is no explanation for whether this assumption makes sense or not for the classes learned by a DNN. In fact, I tend to believe that this assumption is not true at all: if I consider an input that does not lie on the decision boundary of the DNN classifier, then one can go towards all directions of the ambient space and maintain the class label. This implies that the manifold has **the same dimension as the ambient space**, therefore contradicts the paper's assumption.
> >
> > The entire authors response is attacking my statement that the paper assumes each space to be low-dimensional, and the main point of the response is to tell that instead of being *low*-dimensional, they are actually *high*-dimensional. However, `what I really mean by "low-dimensional" is "having a dimension that is lower than the ambient space dimension". I hope this can make my point clear.
> >
> > Thanks

---

> > > ### Author Response · Authors · 2020-11-18
> > > **Thanks for the clarification!**
> > >
> > > Thank you for clarifying your point! You are absolutely correct: for points not on the decision boundary, the class manifold will have the same dimension as the ambient space. However, in practice we observe that the manifold will be extended in some dimensions and extremely thin in other dimensions. We are effectively measuring the number of extended dimensions, which we refer to colloquially as the manifold dimension. Perhaps I can refer you to section 3.3 of our paper where we write,
> > >
> > > """
> > > A hyperplane of dimension $d$ has $d$ infinitely extended directions and $D − d$ directions of size 0. Among the $d$ extended dimensions, we can move arbitrarily far and still find ourselves in the hyperplane. Conversely moving even infinitesimally in the remaining $D−d$ directions will strictly lead to leaving the plane. In our experiments, this will be the case only approximately. In case of class manifolds, no directions will be infinitely thin or long, however, many will be much thinner than others. What we are
> > > measuring is therefore not strictly a dimension (the manifold will always have the full dimension $D$) but rather
> > > effective dimension (Vershynin, 2015). However, our hyperplane model works well in practice.
> > > """
> > >
> > > Of course, a priori it could have been the case that all of the dimensions might have been extended, in which case our cutting planes would always find points that in the class manifold. The fact that we do not always find correctly classified points implies to us that a nontrivial fraction of the dimensions are thin and this has an impact on the behavior of the network.
> > >
> > > I hope this helps to clarify our perspective. If this point is still unclear, we would be more than happy to continue discussing!

---

> > > ### Author Response · Authors · 2020-11-23
> > > **Very similar points addressed in detail in our response to Reviewer 2**
> > >
> > > *In addition to the clarification below, we also responded in detail to similar and other related concerns from Reviewer 2 if you would like to have a look.*
> > >
> > > We would also like to briefly address your comment stating that:
> > > *"if I consider an input that does not lie on the decision boundary of the DNN classifier, then one can go towards all directions of the ambient space and maintain the class label. This implies that the manifold has the same dimension as the ambient space, therefore contradicts the paper's assumption."*
> > >
> > > This is *very easily refuted by the ubiquitous and universal existence of adversarial examples*. If for a point $X$ I can find an $X'$ very close to $X$ but which is classified by a DNN as different class, this means that I left the class manifold $X$ was in somewhere along the way to $X'$. Since adversarial examples can be found *very close* to the original image in the pixel space, the class manifold has to be *very thin* in that direction for me to leave it so quickly. Therefore there definitely do universally exist directions in which class manifolds are very significantly thin, as compared to typical directions.
> > >
> > > Our notion of dimensionality is more similar to the **stable dimension** as defined in *High-Dimensional Probability: An Introduction with Applications in Data Science* [1] using the concept of a Gaussian width of a subset of $\mathbb{R}^D$ rather than the brittle linear algebraic dimension that you might have been imaging. We will make this clearer in the text.
> > >
> > > [1] Roman Vershynin. High-Dimensional Probability: An Introduction with Applications in Data Science. Cambridge University Press, 2018.

---

### Official Review · AnonReviewer4 · 2020-11-08
**Slice, Dice, and Optimize: Measuring the Dimension of Neural Network Class Manifolds**

**Rating:** 6
**Confidence:** 3

**Review:**

This work aims to study the characteristics of the class manifolds provided by a multiclass classifier. In particular, the main goal is to determine the effective dimensionality of these manifolds for the case of neural networks that outputs normalized probabilities. To achieve this, authors introduce the cutting plane method that, following some assumptions, allow them to relate the dimensionality of a random affine hyperplane to the effective dimensionality of the manifold for each class. Authors support their main findings including extensive experimentation.

The theoretical foundation behind the paper seems to be sound, however, as a disclaimer, this research area is not close to the main expertise of this reviewer. In terms of writting, the exposition is clear, however, my main doubt is related to the process to infer the manifold dimensionality for a whole class using a process that depends on each specific instance. It will be good to clarify this point and also the computational complexity involved during the process.

As a recommendation, it will be great to test the method using an artificial case with known ground truth about the effective dimensionality of the class subspaces, so it will be possible to directly validate the findings of the cutting plane method. The current analysis focuses on the case that the cutting plane dimension leads to 50% of the target class (d_50%), which is not the only choice, specially considering that a highly relevant goal is to quantify the effect of manifold dimensionality on generalization. The use of metrics or scores related to generalization will lead to valuable conclusions.

In summary, this is an interesting area of research to shed lights on the process used by learning models to transform from input space to class-probability space. In particular, the potential relation between manifold dimensionality and generalization is worth to pursue. This work will be of interest to ICLR and I recommend to be accepted as a poster contribution.

---

> ### Author Response · Authors · 2020-11-10
> **A reply to Reviewer 4 (1/2)**
>
> Thank you for your detailed review and kind words! We are very happy that you find our experimentation extensive, our theoretical foundations sound, and that you find the exposition clear. We’re also excited that you like the connection of class manifold dimension to generalization we’re uncovering. We hope that you will champion our paper!
>
> We would like to clarify a few points from your review:
>
> ### A single dimension for a large manifold:
> You are right that we infer a single dimension for the whole class manifold (or manifold in between classes) based on more local experiments. There is a fundamental trade off between how local your estimate is and how general it will be. We decided to go for a global estimate, but we have a good reason to believe that the results for each pair (starting image, affine subspace) of a given cutting plane dimension $d$ are consistent with each other and that the global estimate is therefore a good representation of the local estimates.
>
> If you look at our Figures, for example Figure 2 (showing the probability reached vs dimension of the cutting plane $d$ for all 10 CIFAR-10 class manifolds), the individual dots represent single experiments for a specific pair (starting image, affine subspace of dimension $d$). If you look at a particular $x$ axis value = dimension $d$, you will find that there are 10 experiments there (the dots) and that they are broadly in agreement with each other. While it is true that one particular (image,subspace) optimization might reach a higher probability than the other, on average they do seem to behave in a very similar manner, mainly only varying with the dimension $d$ and not with the specific (starting image, affine subspace) choice. This is why we feel justified in estimating a single global number for the manifold since the local measurements end up very consistent with each other. An improved version of this method could estimating local dimensions individually, but that would be beyond the scope of this work and would likely not add much due to the consistency we observe.
>
> ### Time complexity:
> The particular experiment we perform is this: For a particular target class (or a combination of classes to get to a region in between class manifolds), we sweep through a set of cutting plane dimensions $d$ from 1 to the full dimension of the input (3072 for CIFAR-10/100).
>
> For each $d$, we repeat the following experiment several times to get good statistics (typically 10 times): Choose a random starting image $X_0$ that is of a different class than our target class (or any of the target classes for in-between class regions), generate a random orthonormal matrix $M$ of dimension [d,3072], and on the affine subspace specified by ($X_0$, $M$), run Adam for up to 300 steps to find as high a probability of the target class / classes as possible (we actually optimize the cross-entropy loss with that target class vector).
>
> In our published code at https://github.com/anonymous-code-submission/ICLR2021_paper333_anonymous_code_submission you can run this full experiment for a ResNet20v1 on CIFAR-10 in around **0.1 s per affine subspace** ($X_0$, $M$) per dimension $d$ on a **free GPU**. That works out to be **less than 5 minutes** to sweep the dimensions $d$ for each of the 10 CIFAR-10 classes. This is actually a very quick and practical method in to use to diagnose a DNN.
>
> In terms of the computer science (asymptotic) time complexity, the time required to finish our experiment scales with the product of |the number of classes| x |the number of dimensions $d$ to explore| x |the number of repetitions of each experiment| x |the number of steps of the on-affine-space optimization|, so it is linear in each of the variables and therefore very favourable.

---

> ### Author Response · Authors · 2020-11-10
> **A reply to Reviewer 4 (2/2)**
>
> ### An artificial case where $d^*$ is known
> To verify that our method works, we used a very simple toy problem that we present in Figure 4. We defined the class manifold to have dimension n = 900 in a D=1000 dimensional space, and then did our cutting plane experiment exactly as we would do with a real neural network, but now knowing exactly what we are looking for. In Figure 4 you can see that the cutting plane dimension where we are able to hit the 900-dimensional manifold reliably is quite precisely $d^*$ = 100, as we expected from theory. This validates our experimental technique (we tried more combinations of (D,n), but only show as single Figure 4 as they all work out very similarly).
>
> For a more realistic but still a toy case, we could create for example gaussians in the input space corresponding to different classes and have e.g. a linear classifier fit them, and analyze that. However, there the verification would be less clear than for the experiment we did in Figure 4. If you felt this would be helpful for you, we could add this experiment in the Appendix very easily.
>
> ### Looking at other levels beyond d_50%
> To estimate a single dimension for the function probability reached vs cutting plane dimension, we had to choose a somewhat arbitrary threshold of the probability. Essentially, by looking at where the curve crosses p=50%, we are measuring the dimension of the manifold where the class probability is >50%. We chose 50% because it is the threshold where the image is certainly classified as that class (other classes, even if concentrated at a single bin, can only get <50%).
>
> In Figure 9, we do look at d_25%, d_50% and d_75% to see how consistent the results are. Looking at the probability reached vs cutting plane dimension curves, we can see that the larger the probability threshold, the higher the cutting plane dimension needed to hit it, and therefore the lower the dimension of the corresponding manifold. If it helped you with better judging our work, we can extract higher confidence manifold dimensions (e.g. d_90% or d_99%) from some of our experimental results (it’s just a matter of reading off where the curves cross that level) and we could present them in the Appendix. In fact in Figure 11 we also look at d_90%.

---

### Decision · Program_Chairs · 2021-01-07
**Final Decision**

**Decision:**

Reject

**Comment:**

This paper aims to study the dimension of the Class Manifolds (CM) which are defined as the region classified as certain classes by a neural network. The authors develop a method to measure the dimension of CM by generating random linear subspaces and compute the intersection of the linear subspace with CM. All reviewers agree that this is an interesting problem and worth studying.

However, there are major concerns. One question raised by several reviewers is that the goal of this paper is to analyze the dimension of the region that has the same output for the neural network; while the method and analysis are for a single datum. It is not clear if the obtained result is what the paper really aimed at. Another issue is the experimental results are different from that of local analysis. The dimension estimated by using the method in this paper is much higher.

Based on these, I am not able to recommend acceptance. But the authors are highly encouraged to continue this research.